# High-throughput screening of drug libraries identifies a new synergistic drug combination for the treatment of retinoblastoma

Po-Jen Tseng[1]☯, Irina L. Sinenko [1,2]☯, Marc Chambon[3], Damiano Banfi[3],
Santiago Zugbi[4,5], María Belén Cancela[4,5], Kseniya A. Glinkina[1], Gerardo Turcatti[3],
Christina Stathopoulos[2], Paula Schaiquevich[4,5]*, Francis L. Munier[2]*, Paul J. Dyson[1]*

**1** Institute of Chemical Sciences and Engineering, Ecole Polytechnique Fédérale de Lausanne (EPFL), Lausanne, Switzerland, **2** Jules-Gonin Eye Hospital, Fondation Asile des Aveugles, University of Lausanne, Lausanne, Switzerland, **3** Biomolecular Screening Facility, Ecole Polytechnique Fédérale de Lausanne (EPFL), Lausanne, Switzerland, **4** Unit of Innovative Treatments, Hospital de Pediatría JP Garrahan, Buenos Aires, Argentina, **5** National Scientific and Technical Research Council, CONICET, Buenos Aires, Argentina

☯ These authors contributed equally to this work.
* paula.schaiquevich@gmail.com (PS); francis.munier@fa2.ch (FM); paul.dyson@epfl.ch (PD)

## Abstract

Retinoblastoma (RB) management often involves the combination of chemotherapeutic agents (e.g., carboplatin and etoposide) and techniques (e.g., chemotherapy and thermotherapy). Chemoresistance and relapse, as well as systemic and ocular toxicities of current retinoblastoma chemotherapeutics necessitate the implementation of alternative drugs. Fewer resources and concern for pediatric cancer hinder drug discovery and development in an effective way. To overcome the obstacle, a drug repurposing strategy without intuition was adopted to identify promising drug candidates that are both cytotoxic and selective towards human RB Y79 cells *in vitro* and an orthotopic xenograft mice model *in vivo*. By using high-throughput screening, gemcitabine demonstrated high cytotoxicity in Y79 cells, also potentially synergizing with thermotherapy, with minimal impact on a human retinal pigment epithelial RPE-1 cells. Furthermore, the synergistic effect of gemcitabine with carboplatin is superior to the clinically used combination of etoposide with carboplatin. Efficacy studies in orthotopic xenografts showed significant eye survival advantages after intravitreal gemcitabine administration or the combination of intravitreal gemcitabine with systemic carboplatin compared to relevant controls. Importantly, the combination resulted in lower tumor invasion in the optic nerves of the xenografts. Since gemcitabine is an FDA-approved chemotherapeutic agent, already used to treat other pediatric cancers, it could be repurposed to RB treatment alone or in combination with carboplatin, and potentially combined with thermotherapy, providing the basis for an alternative and improved treatment option for RB patients.

**Data availability statement:** All relevant data are within the manuscript and its Supporting Information files.

**Funding:** We thank the Swiss National Science Foundation and EPFL for financial support. This project has also received funding from the Ministry of Education of Taiwan (R.O.C.) and European Union's Horizon 2020 research and innovation program under the Marie Skłodowska-Curie grant agreement No. 754354. • The funders had no role in study design, data collection and analysis, decision to publish, or preparation of the manuscript.

**Competing interests:** The authors have declared that no competing interests exist.

## Introduction

Retinoblastoma (RB) is the most common primary intraocular malignancy of childhood diagnosed in approximately 8,000 children each year worldwide [1]. Patient survival is > 95% in high-income countries, but < 50% globally [2,3]. More than 97.1% of RB tumors form when both *RB1* alleles are mutated in a susceptible cone photoreceptor precursor [4,5]. Loss of the tumor-suppressive functions of the RB protein leads to uncontrolled cell division and recurrent genomic changes during tumor progression [6].

The goal of RB treatment is to save the patient's life, eye(s) and vision. Moreover, preventing metastasis and reducing the risk of long-term secondary tumors, e.g., osteosarcoma, soft-tissue sarcoma, are high priorities in RB management [7,8]. However, the management of RB is complex and involves strategically chosen methods depending on the tumor stage and location [9]. Treatment modalities for RB include focal therapies (e.g., episcleral plaque radiotherapy, cryotherapy, photocoagulation and thermotherapy), chemotherapy and enucleation [10,11].

Nowadays, chemotherapy has become the most common eye-sparing modality, but intravenous chemotherapy is limited by systemic toxicity, drug resistance, and rapid blood clearance [12]. The main hurdle for intravenous chemotherapy in RB is the restricted diffusion of drugs through the blood-retina barrier, limiting the entry of chemotherapeutic agents into the eye and reducing treatment efficacy [13]. Hence, local administration routes of chemotherapy, i.e., intra-arterial, intravitreal and intra-cameral chemotherapy, have emerged as promising methods that favor drug delivery into the eye, and concomitantly avoid deleterious systemic complications [14,15].

Transpupillary thermotherapy with an infrared (810 nm) diode laser is often used alone or, to consolidate chemoreduction, in combination with chemotherapy, specifically with carboplatin [16] to directly heat the tumor to cytotoxic subcoagulation temperatures between 42 and 60 °C. In addition to combining chemotherapy with thermotherapy, the use of drug combinations frequently improves clinical outcome and have multiple advantages compared to monotherapies in cancer. Synergistic drug combinations can offer higher treatment efficacy, often at lower drug doses than those used in single-drug regimens, which may lead to a reduction of adverse side effects [17]. In RB, the main drug combinations used are systemic carboplatin with etoposide with or without vincrsitine [18]. However, the identification of drug combinations that offer high clinical efficacy and low toxicity has often been driven by intuition and experience rather than through systematic investigation [19]. Hence, optimal combinations potentially remain elusive and, to the best of our knowledge, a systematic search for synergistic effect of drug combinations (to be used alone or in combination with thermotherapy) has not been explored in RB treatment. While a systematic approach has been used to identify synergistic pairs involving the nutraceutical resveratrol, no comparable effort has been undertaken for conventional chemotherapeutic agents [20]. Here, we describe a high-throughput screening (HTS) and zero interaction potency (ZIP) method used to repurpose drugs for RB. A library containing 1360 compounds that comprises over 90% clinically approved drugs was investigated to identify drugs that are both active for RB (cytotoxic to Y79 cells) and

selective (non-toxic to non-cancerous retinal pigment epithelial cells, RPE-1). Additionally, the library was investigated in combination with thermotherapy. The overarching goal was to identify candidate drugs that could enhance or complement current chemotherapy regimens, either as monotherapies or in rationally designed combinations, with or without thermotherapy. Furthermore, the mechanism of action of a selected candidate was elucidated *in vitro*, and its efficacy and safety were evaluated *in vivo* following intravitreal administration and compared to current standard-of-care treatments (intravitreal melphalan or systemic combination of etoposide and carboplatin).

## Materials and methods

### Cell culture

Human retinoblastoma Y79 cells and human non-cancerous retinal pigment epithelial RPE-1 cells were purchased from the American Type Culture Collection (ATCC). Y79 cells were cultured in Glutamax RPMI-1640 (Life Technologies, Switzerland) supplemented with 20% (v/v) fetal bovine serum (FBS) (Life Technologies, Switzerland). RPE-1 cells were cultured in Glutamax DMEM medium (Life Technologies, Switzerland) supplemented with 10% FBS. Cells were cultured at 37 °C in a humidified 5% $CO_2$ incubator and subcultured 2–3 times per week. Additionally, RPE-1 cells were harvested with Trypsin 0.05%-EDTA (Life Technologies, Switzerland). Cells used for the assays when cell viability was > 90%.

### Compound libraries

The Prestwick Chemical Library (PCL) was purchased from Prestwick Chemicals (Strasbourg, France) is composed of 1280 chemically and pharmacologically diverse compounds (ca. 90% being FDA-approved drugs). 80 cancer drugs were selected from the Enzo Kinase Inhibitor library and from the École Polytechnique Fédérale de Lausanne, to establish the Collected Drug Library (CDL). All the compounds were stored in the dark at –20 °C under dry air, using an automated storage system. Their chemical integrity was controlled regularly by HPLC-MS. Additionally, carboplatin and gemcitabine were purchased from Tokyo Chemical Industry Co., Ltd (TCI; Tokyo, Japan).

### High-throughput screen and cell viability assay

Compounds were dispensed into sterile barcoded 384-well plates (Corning, Switzerland), using an acoustic liquid handler Echo 550 (Labcyte Inc. Sunnyvale, CA, USA). Plates were prepared in duplicates on which each molecule was tested once. The compounds in the CDL were tested at a fixed concentration of 10 µM. Gambogic acid (GA) was used as a positive control and an equivalent volume of 0.1% DMSO (final concentration) was used as negative control. The 384 well plates used for the assay were prepared by the addition of 30 nl of 10 mM drug to each well. 30 µl of cell suspension was added for the Y79 (4.0•$10^5$ cells/mL) and RPE-1 (1.0•$10^5$ cells/mL). The 384 plates were placed in incubator 37 °C for 72 h treatment time for normal screening. To mimic the hyperthermia condition, the 384 well plates were placed in incubator at 42 °C in the first hour and then transfered to incubator at 37 °C. After total incubation for 72 h, 3 µL of PrestoBlue reagent (Thermo Fisher Scientific, Switzerland) was added to each well and the plates were incubated for 1 h in a cell incubator. The fluorescence intensity (bottom-read) was quantified using a multiwell plate reader (ex560 nm/em590 nm, Tecan Infinite F500). The results from the screens were normalized to the controls for every plate and presented as HTS scores, where a score of 0 corresponds to the average fluorescence intensity of the negative control wells (i.e., no cytotoxic activity) and a score of 1 to that of the positive control wells, and indicates very active compounds. Hit compounds were identified and statistically validated when their HTS scores were higher than the average of the negative controls + 3•SD. The primary scores for the hits were based on the mean from the 4 independent experiments, each performed in duplicate. An in-house Laboratory Information Management System (LIMS) was used for basic data processing, management, visualization, and statistical hit validation. The screening window coefficient (Z'-factor) was used as an indicator for assay development and optimization and as a statistical tool for the assay quality assessment. The Z'-factor was determined as the following: $Z' = 1 - 3 \cdot (SD_{pos} + SD_{neg})/|\mu_{pos} - \mu_{neg}|$, where SD and µ are the standard deviation and the mean,

respectively, of the fluorescence signals of the positive and negative control wells [21]. The quality of the all screened plates was considered sufficient for screening and automation when Z'>0.5.

## Dose-dependent cytotoxicity and synergism assessment

To assess dose-dependent drug responses of the identified hits, Y79 cells were plated at a density of $4.0 \cdot 10^5$ cells/mL, in 384-well plates, and exposed to each compound in the concentration range 0.78 to 100 µM (the range was chosen based on available Rb cytotoxicity data [22–25]. Dose-response curves were generated using a variable slope model with GraphPad Prism 8.1.1. The selected drug candidates were further studied in drug combinations to identify synergies. Cells were then treated with drug combinations over a concentration range 2.5 to 160 µM for carboplatin and 1.66 to 100 µM for second drug to determine the complete combination matrix for antagonism or synergy using the zero interaction potency (ZIP) model. This new model used to estimate the synergy effect has been previously reported [26]. The ZIP method calculates the synergy score for each specific combination concentration, $\delta$, and the average synergy score, $\delta_{avg}$, for the entire drug combination concentration range studied. $\delta > 0$ indicates a synergistic effect and corresponds to the red area in figure whereas $\delta < 0$ relates to an antagonistic effect which is shown in green. $\delta_{avg}$ corresponds to the average synergy in an entire drug combination concentration range rather than one certain point. The initial analysis was performed using the R package SynergyFinder 1.0 (www.synergyfinder.fimm.fi), while statistical significance was assessed through reanalysis with SynergyFinder 2.0.

The effect of carboplatin and gemcitabine, alone and in combination, on cell viability was evaluated, and potential synergistic interactions were assessed using the Excess over Bliss (EOB) algorithm [27,28]. Y79 cells were plated at a density of $4.0 \cdot 10^5$ cells/mL into 96-well plates, and the following day, treated with serial dilutions of carboplatin, gemcitabine, or their combination. After 5 d of treatment, cell viability was measured using the PrestoBlue Cell Viability Assay (Invitrogen). Each condition was assessed in technical triplicates, and the experiment was performed independently three times.

## Analysis of cell death mechanism

The cell apoptotic rate was determined using an Alexa Fluor® 488 annexin V/Dead Cell Apoptosis Kit (Life Technologies) according to the manufacturer's protocol. Briefly, Y79 cells were treated with monodrug (carboplatin, gemcitabine or etoposide), dual drug (carboplatin-gemcitabine, carboplatin-etoposide), or with vehicle as a control, in 6-well plates and incubated at 37 °C for 72 h. After treatment, cells were harvested and washed in cold PBS. After the cell density was determined, cells were diluted in 1x Annexin-binding buffer and stained with Alexa Fluor® 488 annexin V (AV) and propidium iodide (PI) (5 µl of AV and 1 µl of 100 µg/ml PI for each 100 µl of cell suspension). The samples were incubated at room temperature for 15 min, and subsequently analyzed using flow cytometry (CytoFLEX; Beckman Coulter). The apoptotic cells were calculated as the percentage of early apoptosis (FITC Annexin V positive and propidium iodide (PI) negative) and the late apoptosis (FITC Annexin V positive and PI positive). The experiments were performed in triplicates and were repeated 3 times. The data was analyzed using FlowJo software version 10 (FlowJo, LLC).

## EdU incorporation assay

The inhibitory effect of gemcitabine on DNA synthesis was assessed using a Click-iT® EdU (5-ethynyl-2′-deoxyuridine) Cell Proliferation Kit for Imaging, Alexa Fluor™ 488 dye (Life Technologies). Y79 cells ($4.0 \cdot 10^5$ cells/well) were seeded in 96-well plates in triplicate and exposed to gemcitabine (50 nM) or gemcitabine plus carboplatin (50 nM + 50 µM) for 72 h, and then treated with 50 µM of EdU for 2 h at 37 °C. After being fixed with 4% paraformaldehyde for 30 min, cells were transferred to microcentrifuge tubes, treated with 0.1% Triton X-100 for 20 min and washed with phosphate-buffered saline three times. The cells were then exposed to 100 µl of 1x Click-iT ® reaction cocktail for 30 min and incubated with 1x Hoechst 33342 to stain the cell nuclei for 30 min. Cells were gently transferred to glass-bottom imaging dishes for microscopy. Images of the cells were captured with a fluorescence microscope (Leica, Wetzlar, Germany). Fiji software was used to count the fluorescent points [29].

The percentage of EdU incorporation was calculated as the number of EdU-positive nuclei (green) divided by the total nuclei (blue) counterstained with Hoechst 33342.

### Efficacy studies of intravitreal gemcitabine

Animal studies complied with the Statement for the Use of Animals in Ophthalmic and Vision Research of the Association for Research in Vision and Ophthalmology. Approval was granted by the Institutional Animal Care and Use Committee of Fundación Instituto Leloir, Argentina (protocol # 2019−069). The steps to ameliorate suffering included the implementation of the 3Rs principle, developing pain management protocols, and using non-invasive monitoring techniques to minimize distress and improve welfare.

Orthotopic xenografts were established under general anesthesia (ketamine 100 mg/kg and xylazine 10 mg/kg, intra-peritoneally) and topical local anesthesia (0.5% proparacaine hydrochloride ophthalmic solution) after intravitreal injection of $2 \cdot 10^5$ Y79 (HTB-18, ATCC) cells resuspended in 2 µl of Matrigel (BD Bioscience, NJ, USA) into the posterior segment of both eyes of BALBc nu/nu mice as previously described [30,31]. The injections were performed under a stereomicroscope (M80, Leica Microsystems) to ensure accuracy and reduce procedural trauma. Upon completion of the procedure, an erythromycin ophthalmic ointment was applied to prevent corneal desiccation and reduce the risk of infection during the recovery period. After tumor engraftment, mice were randomly divided into 4 groups of 5 animals each and received 2 weekly injections of vehicle (control group) or two cycles of chemotherapy consisting of intravitreal gemcitabine (0.05 ng to attain the $IC_{50}$ value considering 5 µl of vitreous humor in mice), intravitreal melphalan (0.03 µg corresponding to the human-equivalent dose of 30 µg that is the most frequent dose used in the clinics of intraocular retinoblastoma), or intravitreal gemcitabine (0.05 ng) plus systemic carboplatin (34 mg/kg i.p).

Mouse weight and clinical signs of toxicity were monitored daily for each group throughout treatment. Enucleation under anesthesia and analgesia using the protocol described above was performed upon achievement of the experimental endpoint (eyes reaching 3 times the normal size) as previously described [30], and thereafter optic nerves were dissected and stored. If animals were bilaterally enucleated, they were sacrificed by cervical dislocation, brains were collected and both brains and optic nerves were snap frozen and stored at −80 °C. To quantify for human retinoblastoma cell dissemination in the xenograft tissues, the photoreceptor marker cone-rod homeobox RNA was quantified by RT-qPCR using a 7500 Sequence Detection System (Applied Biosystems, Foster city, CA, USA) as detailed elsewhere [31,32].

### Statistical analysis

All experiments were conducted at least in triplicates. Mean and SD values are presented, unless otherwise specified. Statistical analyses for the *in vitro* studies were performed with GraphPad Prism 8.1.1 software. Multiple unpaired t-test was used for pairwise comparisons; two-way ANOVA was used to compare multiple groups. P values < 0.05 were considered statistically significant. A Kaplan Meier eye survival was performed for each group of animals and statistical comparisons between survival curves were performed by means of log-rank test (p < 0.05). Fisher's exact test was used to compare the proportion of optic nerves or brains with positive tumor infiltration between groups. All quantitative data underlying the statistical analyses and figure panels are provided in Supporting Information (S1 Dataset) in S1 File.

## Results

### Primary high-throughput screen of compound collections

The Collected Drug Library (CDL) was used for the initial screen. The library consists of the Prestwick Chemical Library and selected Enzo Kinase Inhibitor library made up of 1360 off-patent, predominantly FDA-approved & EMA-approved drugs. Parallel screens were performed under normal and hyperthermic conditions, the latter to mimic thermotherapy (**Fig 1a**). Cytotoxicity screening was carried out using the CDL in the RB Y79 cell line at a fixed drug concentration of 10 µM with an incubation time of 72 h at 37 °C. HTS scores were calculated by normalizing results to controls for each plate,

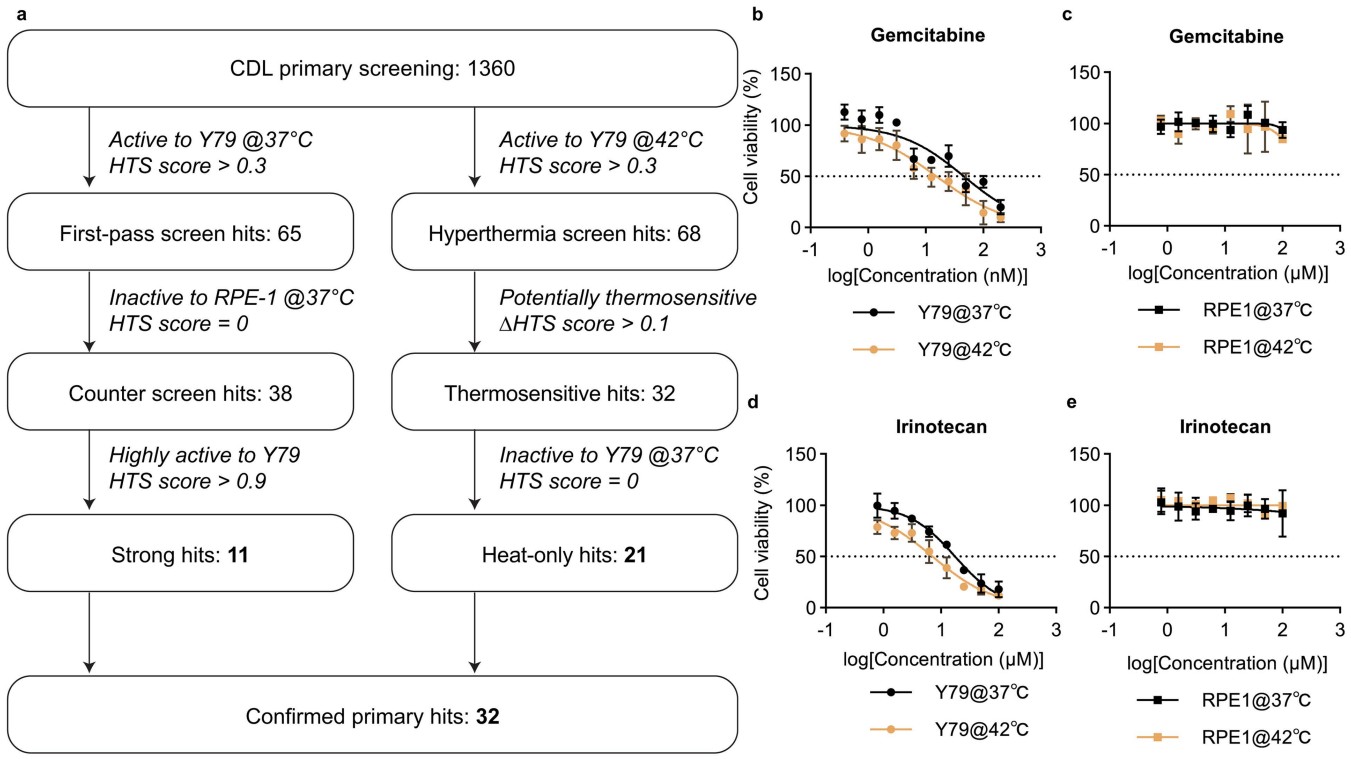

**Fig 1. The Collected Drug Library HTS in RB Y79 and retinal epithelial RPE-1 cells. a Schematic of the primary screening workflow.** Dose-response curves of (b, c) gemcitabine and (d, e) irinotecan in (b, d) Y79 and (c, e) RPE-1 cells at 37 °C (black) and 42°C (yellow). N = 3, mean and SD values are shown.

where a score of 0 represents low cytotoxicity of the negative control, and a score of 1 corresponds to high cytotoxicity of the positive control. Hit compounds were identified when their scores surpassed the average of negative controls plus three times the standard deviation (3•SD). The cytotoxic effect of each drug was represented by a HTS score from 0 to 1, and a drug was classified as "first-pass screen hits" when the HTS score was > 0.3. Drugs with a HTS score < 0.3 have no statistical relevance, and so were assigned with a HTS score of 0. With these criteria, 65 drugs were found to be active against the Y79 cells (**S1 Table**). Subsequently, these first-pass screen hits were tested in non-cancerous retinal pigment epithelial RPE-1 cells, and following this counter screen, drugs with a HTS score lower than the average of the negative (non-treated) control + 3•SD were considered inactive towards the RPE-1 cells and selected for further studies, reducing the number of hits to 38 (**S1 Table**). To further prioritize compounds with strong anti-cancer potential, we applied a data-based selection step by ranking these 38 compounds by their HTS scores in Y79 cells and selecting those with an HTS score > 0.9. This resulted in 11 compounds with the highest cytotoxic activity at 37 °C, which were advanced for further investigation (**Fig 1a**).

Independent screening of the CDL under hyperthermic conditions (i.e., applying an elevated temperature of 42 °C for 1 h at the start of the 72 h incubation to mimic thermotherapy) was performed on Y79 cells [33–35]. 68 hits with a HTS score > 0.3 were identified from this hyperthermia screen (**Fig 1a**). We then aimed to select drugs that are inactive at body temperature but become active at elevated temperatures to mitigate the clinical challenge of chemotherapy-associated toxicity. To recognize the drugs that demonstrated a cytotoxicity enhancement when combined with thermotherapy, ΔHTS score was employed as the difference between HTS scores at 42 °C and at 37 °C, i.e., $HTS_{@42} - HTS_{@37}$ (**S2 Table**). Out

of 32 drugs with ΔHTS score over 0.1 (i.e., with cytotoxicity enhancement with thermotherapy), 21 drugs were inactive at 37 °C (i.e., not identified as hits) and therefore selected for further study.

Taken together,11 drugs with high HTS score > 0.9 obtained at 37 °C and 21 drugs with 1) ΔHTS score > 0.1 and 2) inactive at 37 °C were confirmed as primary hits (S3 Table). These 32 drugs were active against Y79 cells and inactive towards RPE-1 cells. This set includes agents with strong baseline activity at physiological temperature as well as agents whose cytotoxicity is specifically enhanced by mild hyperthermia, indicating complementary routes for potential chemo-therapeutic and chemothermotherapeutic development.

## Secondary confirmation screening and identification of active compounds in Y79 cells

The 32 primary hit drugs underwent secondary confirmation screening, in which active compounds were identified through dose-response assays against Y79 and RPE-1 cells at 37 °C, providing $IC_{50}$ values (S1 and S2 Figs in S1 File). Out of 32 evaluated drugs, 21 demonstrated concentration-response relationships resulting in measurable $IC_{50}$ values in the Y79 cell line, but not in RPE-1 cells (Table 1). 17 drugs have $IC_{50}$ values from 0.04 to 80 µM. The most active drugs in Y79 cells, with $IC_{50}$ values < 10 µM, include benzethonium chloride, chlorhexidine, thonzonium bromide and gemcitabine. Among these, gemcitabine showed the highest cytotoxic potency with an $IC_{50}$ value towards Y79 cells in the nanomolar concentration range (45.97 ± 6.74 nM) (Fig 1b), and was inactive towards RPE-1 cells ($IC_{50}$ > 100 µM) (Fig 1c).

Next, dose-response assays were conducted under hyperthermic conditions. $IC_{50}$ values in the Y79 cell line, and not in RPE-1 cell line, were established for 16 out of 32 primary hits. Out of 32 primary hits, $IC_{50}$ values in Y79 cells ranged from 0.01 µM to 80 µM with only irinotecan and gemcitabine displaying $IC_{50}$ values < 10 µM (Fig 1b and 1d). Notably, the $IC_{50}$ value of irinotecan decreases from 16.94 ± 1.30 µM at 37 °C to 7.11 ± 0.69 µM at 42 °C and for gemcitabine it decreases from 0.045 ± 0.006 µM to 0.015 ± 0.001 µM. A decrease in $IC_{50}$ value obtained at 42 °C suggests the drugs has thermo-sensitive properties that could be advantageous in chemothermotherapy [36,37]. Importantly, both irinotecan and gemcitabine were inactive towards RPE-1 cells at 42 °C ($IC_{50}$ > 100 µM; Fig 1c and 1e).

## Synergistic cytotoxic effect of carboplatin and selected hit drugs

To build on the findings from the single-agent screen and explore clinically actionable regimens, we next investigated rational drug combinations with carboplatin, a part of multi-drug regimens in RB treatment [38]. Four drugs (gemcitabine, irinotecan, aminacrine and trifluridine) from the secondary confirmation screening were selected based on their $IC_{50}$ values and pharmacologic targets that potentially synergize with carboplatin, such as inhibiting DNA synthesis pathways involved in the repair of platinum-DNA adducts [39]. Additionally, etoposide and vincristine, being part of multi-drug protocols for RB treatment, were evaluated under the same conditions.

The cytotoxicity of these 6 drugs (gemcitabine, irinotecan, aminacrine, trifluridine, etoposide and vincristine) was investigated in combination with carboplatin in Y79 and RPE-1 cell lines. The drug combination screening was based on a dose-response matrix design. An 8 x 8 matrix was set up employing 2.5 to 160 µM of carboplatin and 1.66 to 100 µM of each of the selected drugs. Based on cell viability, the synergistic score for each specific combination concentration, $\delta$, and the average synergistic score, $\delta_{avg}$, for the entire drug combination concentration range were calculated using the zero interaction potency (ZIP) method [26] (S4 Table and S3 Fig). $\delta > 0$ indicates a synergistic effect, whereas $\delta < 0$ relates to an antagonistic effect. Discrepancies in absolute cytotoxicity values across experiments likely reflect inherent inter-assay variability, including differences in cell passage, plating uniformity, and positional effects. Importantly, relative response patterns remained internally consistent.

Synergism over the entire concentration range was confirmed for the combinations of carboplatin-gemcitabine, carboplatin-irinotecan, carboplatin-trifluridine and carboplatin-etoposide in Y79 cells, with $\delta_{avg}$ of 18.75, 6.21, 11.80 and 3.45, respectively (Table 2). Carboplatin-aminacrine and carboplatin-vincristine demonstrated $\delta_{avg}$ of −0.24 and −2.98 corresponding to overall antagonism. Interestingly, $\delta_{avg}$ for the combinations of carboplatin-gemcitabine,

**Table 1.  Cytotoxicity (expressed as IC$_{50}$ values) of the 32 primary hit drugs.**

| Drug | IC$_{50}$ [µM] | | | |
|---|---|---|---|---|
| | Y79 | | RPE-1 | |
| | 37 °C | 42 °C | 37 °C | 42 °C |
| Benzethonium chloride | 3.26±0.42 | – | > 100 | – |
| Chlorhexidine | 9.63±1.13 | – | > 100 | – |
| Clofilium tosylate | 19.51±2.12 | – | > 100 | – |
| GBR 12909 dihydrochloride | 24.03±2.95 | – | 31.67±2.76 | – |
| Hexachlorophene | 4.10±1.41 | – | > 100 | – |
| Orphenadrine hydrochloride | > 100 | – | > 100 | – |
| Primaquine diphosphate | 18.29±2.60 | – | 33.58±11.19 | – |
| Retinoic acid | > 100 | – | > 100 | – |
| Thonzonium bromide | 2.71±0.14 | – | > 100 | – |
| Vincamine | > 100 | – | > 100 | – |
| Xylometazoline hydrochloride | > 100 | – | > 100 | – |
| Allantoin | 42.54±4.23 | 37.12±3.46 | > 100 | > 100 |
| Aminacrine | 14.37±2.54 | 10.33±1.21 | > 100 | > 100 |
| Anastrozole | 22.63±1.51 | 21.29±1.15 | > 100 | > 100 |
| Anethole-trithione | 69.59±6.68 | 64.70±4.03 | > 100 | > 100 |
| Argatroban | 81.68±12.35 | 76.26±7.38 | > 100 | > 100 |
| Azaguanine-8 | 32.58±4.65 | 26.09±3.47 | > 100 | > 100 |
| Benzonatate | > 100 | > 100 | > 100 | > 100 |
| Busulfan | 34.07±1.81 | 27.47±1.12 | > 100 | > 100 |
| Cabozantinib | > 100 | > 100 | > 100 | > 100 |
| Carbidopa | 27.92±2.62 | 23.30±1.60 | > 100 | > 100 |
| Carboplatin | 52.94±5.39 | 28.78±4.48 | > 100 | > 100 |
| Ciclesonide | 2.15±0.15 | 2.34±0.23 | > 100 | 90.67±22.83 |
| Docetaxel | > 100 | > 100 | > 100 | > 100 |
| Gemcitabine | 0.045±0.006 | 0.015±0.001 | > 100 | > 100 |
| Irinotecan Hydrochloride | 16.94±1.30 | 7.11±0.69 | > 100 | > 100 |
| Mercaptopurine | 44.51±13.72 | 30.03±7.99 | > 100 | > 100 |
| Oxyphenbutazone | 17.14±1.43 | 19.12±1.07 | > 100 | > 100 |
| Praziquantel | 26.53±4.57 | 29.41±3.57 | 62.12±7.42 | 67.46±5.90 |
| Toremifene | 23.94±1.91 | 20.57±1.41 | > 100 | > 100 |
| Trifluridine | 44.47±4.76 | 20.02±1.99 | > 100 | > 100 |
| Tripelennamine hydrochloride | 48.68±5.10 | 42.63±4.13 | > 100 | > 100 |

carboplatin-irinotecan and carboplatin-trifluridine were superior to clinically used carboplatin-etoposide ($\delta_{avg}$ of 3.45) and carboplatin-vincristine ($\delta_{avg}$ of −2.98). Further analysis using SynergyFinder 2.0 demonstrated that only the carboplatin–gemcitabine combination exhibited statistically significant synergism, with a $\delta_{avg}$ of 8.55 and a p-value of 0.005 (**S5 and S6 Tables**). It was the only drug combination that maintained a favorable therapeutic window at low concentrations.

Among the evaluated drug combinations, carboplatin-gemcitabine exhibited an increase in average synergy score ($\delta_{avg}$) from 18.75 at 37 °C to 27.53 at 42 °C, although this trend did not reach statistical significance (**S6 Table**). Consequently, subsequent investigations focused on evaluating the efficacy of the drug combinations in retinoblastoma models independently of thermotherapy.

**Table 2. ZIP average synergism scores ($\delta_{avg}$) of the selected drug combinations in Y79 and RPE-1 cells under normal and hyperthermic conditions in comparison with the clinically used combinations.**

| Drug combination | ZIP average synergy score/ $\delta_{avg}$ | | | |
|---|---|---|---|---|
| | Y79 | | RPE-1 | |
| | 37 °C | 42 °C | 37 °C | 42 °C |
| carboplatin-gemcitabine | 18.75 | 27.53 | 12.60 | 9.89 |
| carboplatin-irinotecan | 6.21 | 3.62 | 0.56 | 0.98 |
| carboplatin-aminacrine | −0.24 | 0.58 | 1.56 | 2.00 |
| carboplatin-trifluridine | 11.80 | 8.02 | 3.46 | −13.51 |
| carboplatin-etoposide[a] | 3.45 | 1.74 | 11.88 | −2.10 |
| carboplatin-vincristine[a] | −2.98 | −2.66 | −25.21 | 4.02 |

[a]Clinically used drug combinations.

Combining carboplatin with gemcitabine, irinotecan, trifluridine or etoposide resulted in predominantly synergistic interactions over the entire dose-response matrix in Y79 cells (**S3 Fig**). For the carboplatin-gemcitabine combination, the interactions were universally synergistic, and notably, a strong synergistic effect was found even at low concentrations of both drugs (< 5 µM). For example, combining carboplatin (3.3 µM) and gemcitabine (2.5 µM) resulted in 60% cell viability inhibition and a $\delta$ of 18.5 (**Fig 2a**). For carboplatin-irinotecan or carboplatin-trifluridine, synergies were found at higher (> 10 µM) concentrations compared to carboplatin-gemcitabine (**S3 Fig**). To achieve 60% cell viability inhibition, either 12.5 µM of carboplatin and 10 µM of irinotecan ($\delta$ was 5.4), or 20 µM of trifluridine and 20 µM of carboplatin ($\delta$ was 11.2) were required. Similarly, clinically used carboplatin-etoposide showed a strong synergistic cytotoxic effect, but only at higher concentrations (> 10 µM; **Fig 2b**). In contrast, antagonism ($\delta < 0$) was found for the combinations of carboplatin-aminacrine and carboplatin-vincristine (**S3 Fig**).

Drug combination studies on RPE-1 cells under conditions matching those for Y79 cells aimed to determine a therapeutic window, focusing on low concentrations (< 10 µM). Specifically, cytotoxic effects were compared using 3.3 µM carboplatin with 2.5 µM of gemcitabine, irinotecan, aminacrine, trifluridine, etoposide or vincristine (**Fig 2c**). At these doses, carboplatin-gemcitabine resulted in 60% cell viability inhibition in Y79 cells compared to 20% in RPE-1 cells, indicating a promising therapeutic window. In contrast, carboplatin-irinotecan resulted in 25% cell viability inhibition in Y79 cells and 18% cell viability inhibition in RPE-1 cells. The carboplatin-trifluridine combination resulted in 30% cell viability inhibition in both Y79 and RPE-1 cells. Interestingly, the clinically used carboplatin-etoposide combination showed low cell viability inhibition (< 20%) in both cell lines and a weak synergism ($\delta < 10$). This combination is known for its schedule- and exposure-dependent activity, achieving maximal efficacy with sequential (carboplatin-before-etoposide) and repeated dosing rather than the single-time, concurrent low-dose exposure used in our assay [40,41]. While both carboplatin-gemcitabine and carboplatin-vincristine resulted in 60% cell viability inhibition in Y79 cells, carboplatin-vincristine demonstrated a strong antagonistic effect ($\delta < -10$).

Next, a feasible therapeutic window was investigated for low concentration range of carboplatin-gemcitabine (1 µM and 0–100 nM, respectively) (**Fig 2d**). A reduction of cell viability (from 86 to 17%) in Y79 cells was observed for the carboplatin-gemcitabine combination, importantly with no apparent cytotoxicity observed in RPE-1 cells when exposed to the combination in the same concentration range. Next, a 5 x 5 matrix was used to determine the efficacy of low doses of gemcitabine, i.e., < 100 nM, when applied in combination with carboplatin. The combination of carboplatin and gemcitabine significantly increased cytotoxicity in Y79 cells compared to either drug alone, particularly at higher carboplatin concentrations (25–100 µM) across multiple gemcitabine doses (12.5–100 nM; **Fig 2e**). ZIP analysis confirmed that the interaction between carboplatin and gemcitabine was universally synergistic with a $\delta_{avg}$ value of 11.78 (**Fig 2f**).

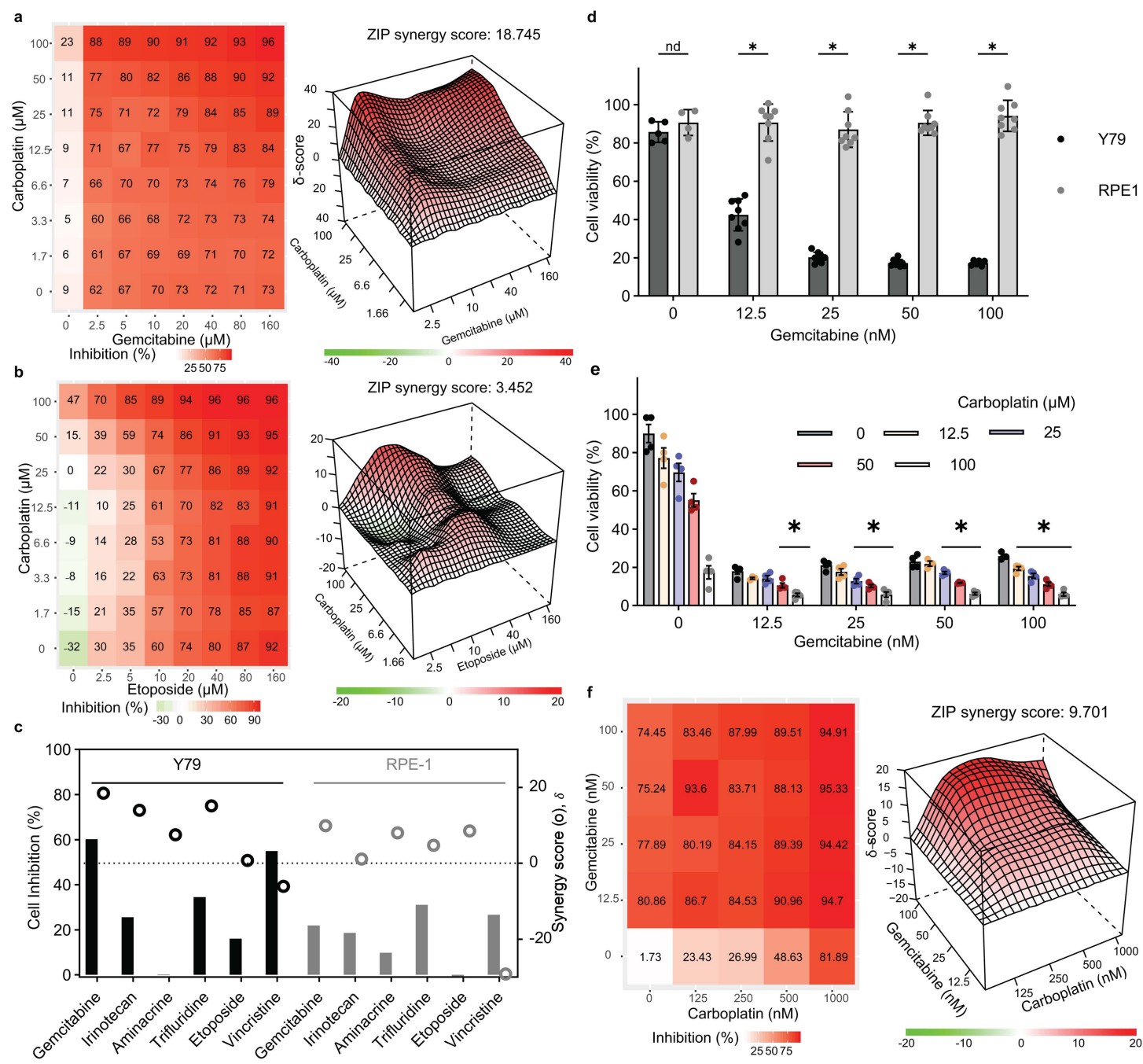

**Fig 2. Identification of the most potent drug combination for the treatment of RB.** Heat maps of cell viability inhibition (left) and ZIP synergy scores (right) for the combinations of carboplatin with (**a**) gemcitabine and (**b**) etoposide in Y79 cells. **c** Cell viability inhibition (bars) and ZIP synergy scores (dots) for the combination of 3.3 μM carboplatin and 2.5 μM of either gemcitabine, irinotecan, aminacrine, trifluridine, etoposide or vincristine in Y79 and RPE-1 cells. **d** Cell viability of 1 μM carboplatin with 0–100 nM gemcitabine in Y79 and RPE-1 cells; unpaired t-test. **e** Cell viability of 0 - 100 μM carboplatin with 0–100 nM gemcitabine in Y79 cells; two-way ANOVA with Dunnett test. **f** Heat maps of cell viability inhibition (left) and ZIP synergy scores (right) for the combination of 0–1000 nM carboplatin and 0–100 nM gemcitabine in Y79 cells. N = 3, p < 0.05(*), mean and SD values are shown.

Additionally, the putative synergistic effect of the carboplatin–gemcitabine combination was evaluated using the Excess over Bliss algorithm (**S4 Fig**). In the Y79 cell line, a significantly greater cytotoxic effect was observed with the combination treatment compared to either single agent, and positive Excess over Bliss scores indicated synergism with the maximum value of 22.3 for carboplatin-gemcitabine (2.5 µM and 2.5 nM, respectively). In contrast, no significant additive effect was observed in RPE-1 cells, and the scores were sub-zero and negative (score values of −8.0–0.02), indicating a lack of synergy.

## Combining carboplatin with gemcitabine leads to inhibition of Y79 cell proliferation, increased apoptosis, and inhibition of DNA synthesis

A comparison of the efficacy of the carboplatin-gemcitabine and carboplatin-etoposide combinations, as well as the individual drugs at the concentrations corresponding to their $IC_{50}$ values (50 µM carboplatin, 50 nM gemcitabine or 1 µM etoposide) was assessed in Y79 cells (**S5 Fig**). Carboplatin alone led to a cell viability of 56%, whereas the carboplatin-gemcitabine combination resulted in 12% viable cells. In comparison, the viability of Y79 cells treated with carboplatin-etoposide decreased to 47% compared to 60% with etoposide alone, thus having a lower effectiveness than the carboplatin-gemcitabine combination.

Furthermore, induction of apoptosis by the drug combinations was assessed by Annexin V staining (**S6 Fig**). In the samples treated with either carboplatin, gemcitabine or etoposide alone the fraction of early apoptotic cells was 26, 54 and 49%, respectively. The number of early apoptotic cells in the samples treated with either carboplatin-gemcitabine (66%), or carboplatin-etoposide (59%) was higher than either of the single treatments; however, the proportion of late apoptotic cells was lower in combinations.

Since carboplatin is an alkylating agent causing DNA damage [14], and gemcitabine is a nucleoside analogue that induces cell death via the inhibition of DNA synthesis [33], inhibition of DNA synthesis triggered by gemcitabine alone or combination with carboplatin was assessed using the 5-ethynyl-2′-deoxyuridine (EdU) incorporation assay (**Fig 3a and** 3b). A reduction in EdU-incorporation was observed upon treatment with gemcitabine, decreasing from 35% in untreated cells to 24%, confirming the inhibition of DNA synthesis. Carboplatin alone resulted in 32% EdU incorporation. The carboplatin-gemcitabine combination further reduced EdU-incorporation to 14%, indicating a potentiated effect. One-way ANOVA did not reveal a statistically significant difference across all treatment groups (p = 0.0692 for carboplatin-gemcitabin vs non-treated control); however, a subsequent unpaired t-test showed that EdU incorporation in the carboplatin–gemcitabine group was significantly lower compared to the untreated control (p = 0.0291). These findings suggest that carboplatin may act as an adjuvant by enhancing the inhibitory effect of gemcitabine on DNA synthesis.

## Intravitreal gemcitabine delays tumor progression in orthotopic xenografts

A previously established and fully characterized orthotopic xenograft model [31] was used to compare the efficacy of intravitreal gemcitabine and the combination of systemic carboplatin plus intravitreal gemcitabine to that attained with intravitreal melphalan (as the standard-of-care for intraocular retinoblastoma). All groups of animals were also compared to vehicle-treated eyes as controls (**S7 Fig**). Given that retinoblastoma dissemination primarily occurs locally via the optic nerve [42], and prior studies (including ours) have not detected systemic spread within the experimental timeframe [43], each eye and optic nerve was considered an independent unit of analysis. This approach is further supported by clinical evidence indicating that bilateral tumors in patients can behave independently [44,45].

Doses and treatments were well tolerated by the animals. The median (range) eye survival of animals treated with vehicle, gemcitabine, carboplatin plus gemcitabine, or melphalan, was 31 days [23–38], 37 days [34–49], 41 days [35–49] and 40 days [36–51], respectively. Thus, a significant increase in eye survival was observed after treatment with gemcitabine, melphalan, and the carboplatin-gemcitabine combination, with respect to vehicle-treated animals (log-rank test, p < 0.05, **Fig 4a**). Furthermore, a trend towards higher eye survival was detected in animals treated with carboplatin plus

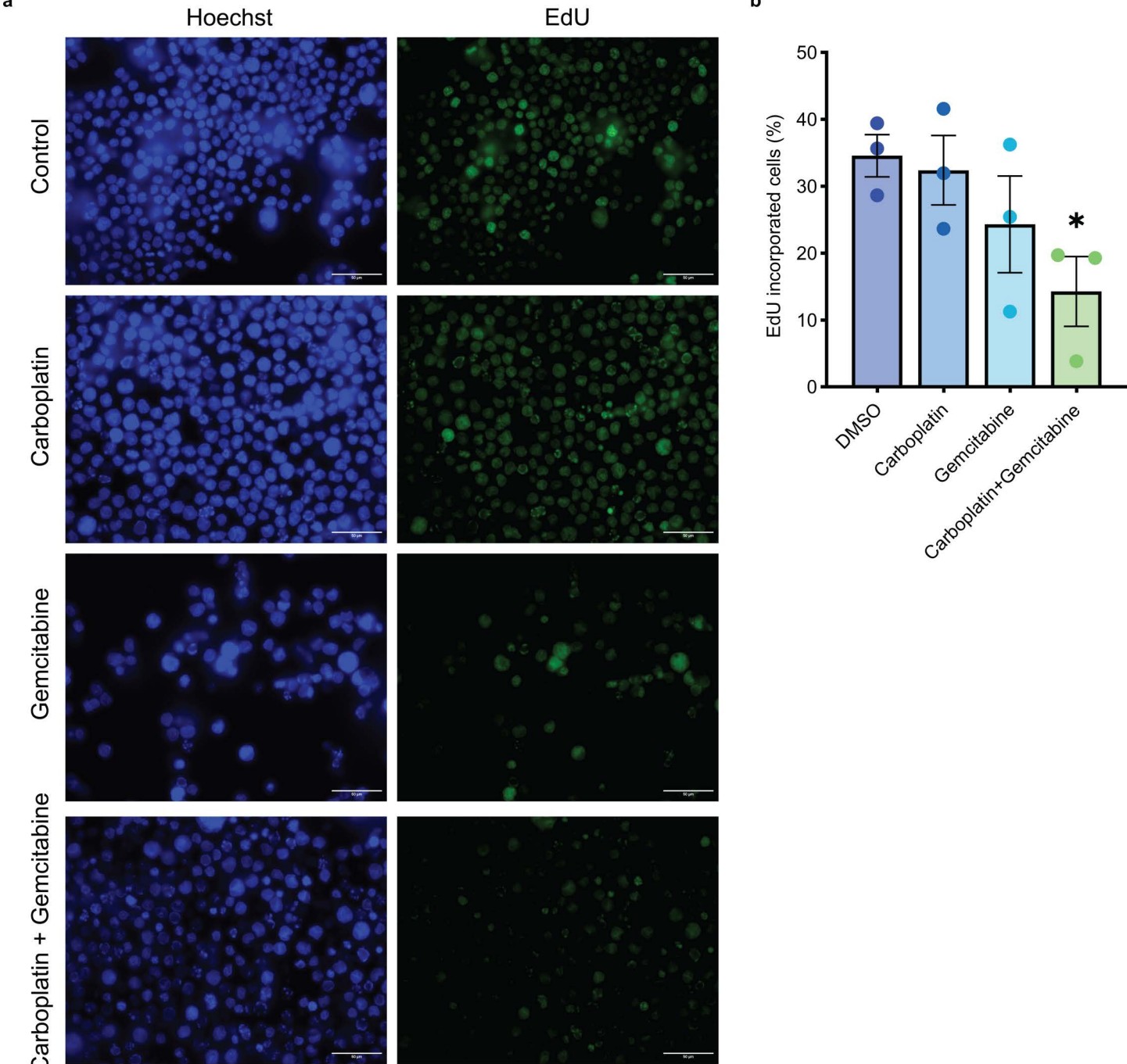

**Fig 3. Effects of gemcitabine and carboplatin on DNA synthesis. A Inhibition of DNA synthesis determined using the EdU incorporation assay for gemcitabine and carboplatin alone and their combination, cell nuclei (blue) and EdU (green) are shown, scale bars 100 μm, representative images of three independent experiments are shown. b Quantification of the EdU incorporation, more than 100 cells per condition were analyzed using Fiji software.** Mean and SEM are shown, N = 3, unpaired t-test, $p < 0.05$ (*).

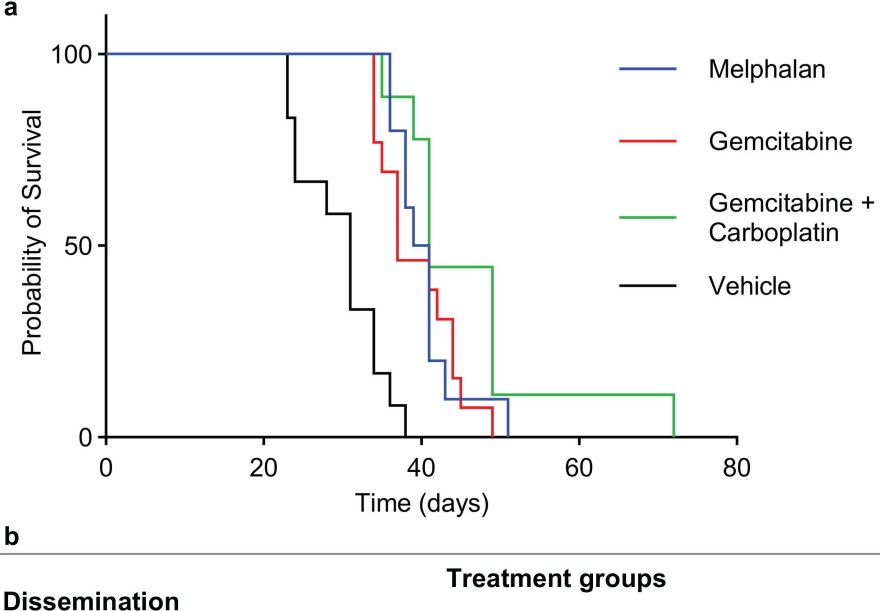

**a**

**b**

| Dissemination site | Treatment groups | | | |
|---|---|---|---|---|
| | Vehicle | Gemcitabine | Gemcitabine + Carboplatin | Melphalan |
| **Brain** | 3/5 | 3/5 | 2/5 | 4/5 |
| **Optic nerve** | 10/10 | 8/10 | 4/10 | 9/10 |

**Fig 4. Efficacy of gemcitabine and carboplatin-gemcitabine. a Eye survival curves of Y79 xenografts after intravitreal melphalan (blue), intravitreal gemcitabine (red), intravitreal gemcitabine plus systemic carboplatin (orange), and vehicle (black). b Proportion of tissues infiltrated with retinoblastoma cells according to treatment groups.** A total of 5 animals (5 brains, 10 optic nerves) were analyzed per treatment group.

gemcitabine compared to those that received gemcitabine as a single-drug (log-rank test, p = 0.06). No significant differences in eye survival were observed between groups of animals treated with gemcitabine and melphalan (log-rank test, p = 0.8), and between animals that received carboplatin plus gemcitabine versus those treated with melphalan (log-rank test, p = 0.1).

Tumor dissemination was detected in 60% of the brains and in 100% of the optic nerves of the vehicle group in accordance with the xenograft characteristics previously reported [31]. In chemotherapy-treated animals, molecular detection of retinoblastoma cells yielded positive results in 60, 40 and 80% of the brains and 80, 40 and 90% of the optics nerves of the animals treated with gemcitabine, carboplatin plus gemcitabine, or melphalan, respectively (**Fig 4b**). Therefore, in addition to a trend towards increased ocular survival, the combination of systemic chemotherapy with carboplatin plus gemcitabine resulted in a reduction in optic nerve invasion of retinoblastoma cells (Fisher's exact test, p = 0.019). Comparison of the effect the different chemotherapy treatments on brain tumor dissemination did not yield significance probably due to the limited number of animals treated in each group.

## Discussion

Chemotherapeutics play a key role in RB management, and while in high-income countries the success rate of treatments is extremely high, chemotherapy-related side effects, chemoresistance and relapse remain a significant burden. With the

goal of finding superior treatments for RB, ideally chemotherapeutics with increased efficacy and fewer side effects, an *in vitro* HTS of a library of 1360 drugs was undertaken. The HTS was designed to identify drugs that (1) exhibit an anti-proliferative effect specifically in RB cells, while not impacting normal retinal pigment epithelial cells, and (2) potentially synergize with thermotherapy. 32 hits were identified based on the primary screening performed at a fixed single dose (**S3 Table**). Next, dose-response relationships of each hit revealed 4 drugs with optimal cytotoxicity profiles ($IC_{50}$ < 10 µM in Y79 cells) and cancer cell selectivity ($IC_{50}$ > 100 µM in RPE-1 cells). Gemcitabine (2'deoxy-2'2'-difluorocytidine), an active nucleoside analogue [46], was identified as the most promising drug, exhibiting activity in the nanomolar range.

Notably, gemcitabine is employed in the treatment of non-small cell lung [47], pancreatic [48], bladder [49], and breast [50] adult cancers, but also pediatric non-Hodgkin lymphoma [51,52], sarcoma [53], and hepatocellular carcinoma [54], and is not currently used in RB treatment. Multi-drug regimens often demonstrate greater efficacy and in certain instances lower toxicity, compared to single-drug regimens [55]. However, there are no strict criteria to define effective combination therapies [56], although gemcitabine and platinum-based drug combinations stand out as highly effective treatments for various cancers, demonstrating notable synergistic effects [57–60]. For instance, studies in phase III trials for advanced non-small-cell lung cancer have demonstrated that carboplatin-gemcitabine regimens exhibit comparable activity to other treatment regimens [61]. Another nucleoside analog, trifluridine, currently used as a topical treatment for viral eye infections [62], incorporates into DNA to inhibit cell proliferation [55]. Recently, trifluridine combined with tipiracil, a thymidine phosphorylase inhibitor used to increase trifluridine bioavailability, was used to treat colorectal [63] and gastric [64,65] cancers.

Inhibitors of topoisomerase I and II, enzymes involved in DNA replication, transcription, chromosome segregation and recombination [66], are used as anti-cancer agents, including in combination with platinum-based drugs. The carboplatin-etoposide combination (note that etoposide is a topoisomerase II inhibitor) is widely employed in RB management [15]. Irinotecan, a topoisomerase I inhibitor, is used to treat colorectal cancer [67,68] and an carboplatin-irinotecan combination demonstrated prolonged survival in extensive-disease small-cell lung cancer [69]. Furthermore, a liposome-encapsulated formulation of irinotecan Onivyde®, was approved for use in patients with metastatic pancreatic adenocarcinoma [70].

Hence, a systematic investigation of carboplatin combined with gemcitabine, irinotecan, aminacrine, trifluridine, etoposide and vincristine, was conducted using a dose-response matrix. The ZIP method, which is based on the hypothesis that two non-interacting drugs will simply affect their dose–response curves, but not the half maximal inhibitory concentration ($IC_{50}$) or the shape of the curve, was employed to determine synergistic behavior *in vitro*. The method is advantageous compared to identifying synergism or antagonism using a combination index based on a single concentration combination, as it takes into account a broader range of concentration combinations. The interaction maps in Y79 and RPE-1 cells reveal that combining carboplatin and gemcitabine yields greater efficacy than either drug alone. Moreover, at low concentrations, carboplatin-gemcitabine display selectivity for RB cells over non-cancerous retinal pigment epithelial cells, i.e., 1 µM carboplatin and < 100 nM gemcitabine is highly cytotoxic to Y79 cells with no apparent cytotoxicity observed in RPE-1 cells. The broad therapeutic window is beneficial for translating this combination into preclinical models, and the prominent synergistic cytotoxicity of carboplatin-gemcitabine towards Y79 cells, specifically at low concentrations, highlights the potency of the combination for the treatment of RB. By using lower doses, the combination of drugs may boost effectiveness while mitigating chemotherapy-related side effects [19]. In comparison to current clinically employed combinations like carboplatin-etoposide and carboplatin-vincristine, the carboplatin-gemcitabine combination demonstrated a superior cytotoxic effect with robust synergistic interactions and the potential to synergize with thermotherapy. Increased levels of apoptotic cell death and DNA synthesis inhibition further validate the potential benefits of the carboplatin-gemcitabine combination compared to either drug alone and other combinations.

Our approach leveraged high-throughput screening on the most aggressive commercially available retinoblastoma cell line, representing later disease stages, and included only one type of non-cancerous retinal cells, and, therefore, may not fully capture the heterogeneity of retinoblastoma, particularly across different molecular subtypes. Expanding the pool

of disease models, including patient-derived or genetically diverse cell lines and *in vivo* systems, would help to address these limitations and enhance the clinical relevance of our findings.

The carboplatin-gemcitabine combination displayed extended eye survival compared to non-treated control, as well as a reduction in the optic nerve tumor invasion in Y79 orthotopic xenografts compared to single-drug treated eyes, supporting its potential for clinical translation. This survival benefit is consistent with previous findings from a murine retinoblastoma model, where a carboplatin–docetaxel combination resulted in significantly prolonged survival compared to either agent alone [71]. Hence, carboplatin plus gemcitabine could offer a novel option for RB treatment in relapsing cases refractory to all available agents, especially for relapsing intraocular and non-CNS metastatic disease.

A future direction of this study involves evaluating the combined effect of gemcitabine and carboplatin with the addition of thermotherapy. While preclinical models of transpupillary thermotherapy (TTT) for retinoblastoma are primarily established in larger animals, such as rabbits [72,73], they provide a viable platform for investigating the potential synergistic effects of this treatment combination. Clinically, gemcitabine-based regimens combined with hyperthermia have shown feasibility in patients with pancreatic adenocarcinoma and have demonstrated a trend toward improved overall survival [58,74,75], supporting the rationale for exploring this approach in ocular oncology models.

To provide a preliminary reference point for potential clinical translation, we compared the intravitreal dose of gemcitabine used in our preclinical model to that of melphalan, the current standard for intravitreal chemotherapy in retinoblastoma, which is typically administered at doses of 20–30 μg [76]. In the mouse experiments, 0.05 ng of gemcitabine was injected into an eye with a vitreous volume of approximately 5 μL. Assuming a vitreous volume of 3.4 mL for a 5-year-old child, based on ocular growth data [77,78], this would be expected scale to an estimated human-equivalent dose of ~34 ng. While this extrapolation is based solely on proportional volume scaling and does not account for interspecies differences in drug distribution, clearance, or retinal penetration, it suggests that gemcitabine may lower intravitreal doses compared to current agents.

Given that standard intravitreal therapies have shown high efficacy in managing vitreous seeds, we do not propose gemcitabine as a replacement in frontline treatment. Instead, its role may potentially lie in refractory intraocular retinoblastoma, where conventional options are limited, or in metastatic disease through systemic or intra-arterial routes. Notably, both gemcitabine and its combination with platinum-based drugs have been used clinically in pediatric oncology [79,80], which may support further investigation of this approach. Nonetheless, these findings are exploratory and require substantial additional validation, including safety profiling and pharmacokinetic studies, before consideration for clinical application.

The safety profile of the gemcitabine–carboplatin combination has been reported across various solid tumors, with hematologic toxicities, such as grade 3/4 neutropenia and thrombocytopenia, being the primary concerns [81–84]. In retinoblastoma, severe hematologic events are documented with both intra-arterial and systemic chemotherapy and require close clinical monitoring and, when indicated, dose modification [85,86]. In the pediatric population, toxicity data on this combination remain scarce. Notably, gemcitabine-based regimens, such as those combining gemcitabine with oxaliplatin or irinotecan, have demonstrated a tolerable safety profile in children [79,80]. Additionally, intra-arterial delivery may offer a strategy to minimize systemic toxicity in retinoblastoma patients, potentially improving the therapeutic index of this approach [87].

In conclusion, this study aimed to address the clinical challenge of limited treatment options for retinoblastoma by employing a systematic approach to identify effective therapeutic strategies. Through high-throughput screening of 1,360 compounds, we evaluated single agents, drug combinations, and the integration of chemotherapy with thermotherapy, modalities widely used in retinoblastoma treatment. Our findings highlight the gemcitabine-carboplatin combination as a promising candidate, demonstrating efficacy comparable to standard-of-care treatments in preclinical models. While further investigation is required to assess its clinical applicability, this study provides a foundation for expanding the therapeutic landscape for retinoblastoma, with the potential to improve treatment outcomes for patients with refractory or relapsed disease.

 

## Supporting information

**S1 File. Dose-response curves for 11 hit compounds in Y79 and RPE1 cells at 37°C.**
(ZIP)

## Acknowledgments

The authors thank the Biomolecular Screening Facility (BSF) of the Ecole Polytechnique Fédérale de Lausanne, specifically Nathalie Ballanfat and Julien Bortoli, for technical assistance in cell culture and HTS automation.

## Author contributions

**Conceptualization:** Paul J. Dyson.

**Formal analysis:** Po-Jen Tseng, Irina L. Sinenko, Damiano Banfi, Santiago Zugbi, Kseniya A. Glinkina.

**Funding acquisition:** Paul J. Dyson.

**Investigation:** Po-Jen Tseng, Irina L. Sinenko, Santiago Zugbi, María Belén Cancela, Kseniya A. Glinkina.

**Methodology:** Po-Jen Tseng, Irina L. Sinenko, Marc Chambon, Damiano Banfi, Santiago Zugbi, María Belén Cancela.

**Project administration:** Gerardo Turcatti, Paula Schaiquevich, Francis L. Munier, Paul J. Dyson.

**Resources:** Paul J. Dyson.

**Supervision:** Paula Schaiquevich, Francis L. Munier, Paul J. Dyson.

**Validation:** Po-Jen Tseng, Irina L. Sinenko, Marc Chambon.

**Visualization:** Irina L. Sinenko, Damiano Banfi.

**Writing – original draft:** Po-Jen Tseng, Irina L. Sinenko, Paul J. Dyson.

**Writing – review & editing:** Po-Jen Tseng, Irina L. Sinenko, Marc Chambon, Damiano Banfi, Santiago Zugbi, María Belén Cancela, Kseniya A. Glinkina, Gerardo Turcatti, Christina Stathopoulos, Paula Schaiquevich, Francis L. Munier, Paul J. Dyson.

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
