## [Decision Letter · Decision Letter 0]

26 Dec 2024

Dear Dr. Dyson,

Thank you for submitting your manuscript to PLOS ONE. After careful consideration, we feel that it has merit but does not fully meet PLOS ONE’s publication criteria as it currently stands. Therefore, we invite you to submit a revised version of the manuscript that addresses the points raised during the review process.

Dear Dr. Paul J. Dyson,

Following the review of your Research Article titled " High-throughput screening of drug libraries identifies a new synergistic drug combination for the treatment of retinoblastoma", I recommend that it should be revised taking into account the changes requested by the reviewers. Since the requested changes are major, the revised manuscript will undergo a second round of review by the same reviewers.

We look forward to receiving your revised manuscript.

Kind regards,

Mounir Tilaoui, Ph.D

Academic Editor

PLOS ONE

3. Thank you for stating the following financial disclosure:  [We thank the Swiss National Science Foundation and EPFL for financial support. This project has also received funding from the Ministry of Education of Taiwan (R.O.C.) and European Union’s Horizon 2020 research and innovation program under the Marie Skłodowska-Curie grant agreement No. 754354.].  Please state what role the funders took in the study.  If the funders had no role, please state: "The funders had no role in study design, data collection and analysis, decision to publish, or preparation of the manuscript." If this statement is not correct you must amend it as needed. Please include this amended Role of Funder statement in your cover letter; we will change the online submission form on your behalf.

4. We note that your Data Availability Statement is currently as follows: [All relevant data are within the manuscript and its Supporting Information files.] Please confirm at this time whether or not your submission contains all raw data required to replicate the results of your study. Authors must share the “minimal data set” for their submission. PLOS defines the minimal data set to consist of the data required to replicate all study findings reported in the article, as well as related metadata and methods (https://journals.plos.org/plosone/s/data-availability#loc-minimal-data-set-definition).

6. Please include a separate caption for each figure in your manuscript.

Additional Editor Comments (if provided):

Reviewers' comments:

Reviewer's Responses to Questions

**Comments to the Author**

1. Is the manuscript technically sound, and do the data support the conclusions?

Reviewer #1: Partly

Reviewer #2: Yes

Reviewer #3: Partly

Reviewer #4: Yes

Reviewer #5: Yes

2. Has the statistical analysis been performed appropriately and rigorously?

Reviewer #1: Yes

Reviewer #2: I Don't Know

Reviewer #3: No

Reviewer #4: Yes

Reviewer #5: Yes

3. Have the authors made all data underlying the findings in their manuscript fully available?

Reviewer #1: Yes

Reviewer #2: No

Reviewer #3: Yes

Reviewer #4: Yes

Reviewer #5: Yes

4. Is the manuscript presented in an intelligible fashion and written in standard English?

Reviewer #1: Yes

Reviewer #2: Yes

Reviewer #3: Yes

Reviewer #4: Yes

Reviewer #5: Yes

Reviewer #1: Page 12, line 267 – For the 21 drugs advanced that showed enhancement with thermotherapy, why did the authors choose only drugs that were not active at 37oC? Wouldn’t it be more advantageous to have a drug that both works at normal body temperature and then is further enhanced with thermotherapy?

Page 15, line 326 – I would recommend adding the avg values for carboplatin-etoposide and carboplatin-vincristine in the text as well.

Page 15, Line 341 – authors state that the combination of carboplatin (3.3 uM) and gemcitabine (2.5 uM) only resulted in 60% cell inhibition. Yet, the IC50 of gemcitabine alone is listed as 0.045 uM. Similarly, the dose-response curve in Figure 1 for gemcitabine shows ~60% inhibition at concentrations of 0.1 uM. If the curve in Figure 1 is to be believed, gemcitabine at 2500 nM (Log ~3.4) would be expected to be close to 100% inhibition on its own. Why the big discrepancy in cell viability/% inhibition values between Figure 1 and Figure 2?

Page 15, Line 342 – Quantitating synergistic effects is not trivial to perform and even more difficult to describe. For example, the line “This corresponds to a 15-fold reduction in the concentration of carboplatin required to produce such an effect when compared to carboplatin alone”, yet according to Figure 2, this same % inhibition (~60%) is observed even when 0 uM of carboplatin is used. It is also unclear where the 15-fold reduction number comes from as 15 fold of 3.3 uM is ~50uM and yet 50uM in Figure 2a with no Gemcitabine only inhibited ~11%.

Page 16, line 368 – Authors state “At these doses, carboplatin-gemcitabine resulted in 60% cell viability inhibition in Y79 cells compared to 20% in RPE-1 cells, indicating a promising therapeutic window.” However, the numbers they quote do not appear to be what is depicted in Figure 2c, carboplatin-gemcitabine in Y79 cells looks to be ~60% inhibition as they state, but in RPE-1 cells, the inhibition looks to be even higher inhibition ~65%. No ideas where authors get 20% cell inhibition value from. The figure in 2c makes it appear that there is no therapeutic window using any of the chemotherapeutics as almost all Cell inhibition is similar between Y79 and RPE-1 cells.

Similarly, the other cell inhibition percentages for the other drugs do not appear to be correct either. Authors state 25% and 18% inhibition for carboplatin-irinotecan in Y79 and RPE-1 cells, respectively. However, the graph in Figure 2c clearly shows a greater amount of cell inhibition for the RPE-1 cells.

Page 18, line 397 – Authors state “Furthermore, drug combinations induced greater levels of apoptosis than single 398 drugs as determined by flow cytometry (Fig. 3a,b).” Yet, the flow graphs in Figure 3a do not back up this statement. In fact, Gemcitabine alone had 54.2% in the early apoptotic region and 19.2% in the late apoptotic/dead cell region. Yet the carboplatin + gemcitabine only had 46.9% in early and 12.4% in late. Similarly, etoposide alone appeared to do better in apoptotic induction as compared to the carboplatin + etoposide.

For the quantitation in Figure 2b, were the graphs in figure 2a not used for the analysis? I assume the dots on the bar chart are the individual data points. So if we take etoposide as an example, why is there no dot at 66% (or 76% if you are combining early and late apoptosis regions)?

Reviewer #2: This manuscript present the high-throughput screening of a drug library to identify candidates for synergistic drug combinations for the treatment of retinoblastoma. The study is straightforward and clearly described and the manuscript is well written. However, I believe there are some (minor) issues that need to be addressed by the authors.

Aim of library screening

In the introduction, the authors clearly describe the goal of the high-throughput screening, to identify candidates that are both active for RB (cytotoxic to Y79 cells) and selective (non-toxic to RPE-1) with and without thermotherapy. However, after identifying candidates through the high-throughput screen, candidates are tested in drug combinations and the focus shifts towards multi-drug treatments and away from thermotherapy. If the goal is to identify candidates for multi-drug treatments, why not include combinations with carboplatin in the initial screen? And why use the delta HTS parameter for score if the downstream combination with carboplatin are performed at 37C? Also, authors introduce the drug library as a general drug library of approved drugs. But during hit selection, pharmacologic targets (line 305) are used as a criteria. So, why not only screen drugs with relevant pharmacologic targets? Or why exclude hits with no relevant target but identified through the screen? And it was little disappointing to found out at the end of the discussion that gemcitabine and platinum-based drug combinations are already used as effective treatments. The authors should address these issues and explain in more detail why certain choices were made. Furthermore, the authors should explain what the third round of screening entailed and why it was performed. And the scheme

Minor issues

- Line 170-171: Sentence repeated

- Line 237: Approved by who? Add to text

- Line 241-242: Please explain the scoring the main text. It is explained in the methods section, but this is important for the results section.

- Line 325: Are the authors comparing the observed in vitro screening results with other in vitro results? If so, please add reference. If there refer to actual clinical data, the authors should not compare in vitro screening data to clinical treatments.

- Line 533-549: A lot of speculation. The authors should tone down this section.

- I'm missing a conclusion section or conclusion paragraph at the end of the manuscript

Figures

- Figure 1b: Not informative, as selections can be stated as text

- Figure 3. FACS scatter plots look all very similar and numbers are not readable. Overall not informative. Edu images are counter to the bar plot, as it looks like, there is more fluorescent signal in the gemcitabine treated cells

Reviewer #3: Tseng et al. describe the results of a high-throughput screening campaign in retinoblastoma cells (Y79) for the purpose of identifying synergistic drug combinations and synergy with thermotherapy. The combination the authors focused on post-screening was carboplatin with gemcitabine. This combination demonstrated a favorable therapeutic window, reducing the required carboplatin dose by 15-fold while maintaining cytotoxicity in Y79 cells and sparing RPE-1 cells. The authors also perform mechanistic studies. In vivo studies using an orthotopic xenograft model of RB showed that intravitreal gemcitabine, alone or with systemic carboplatin, delayed tumor progression compared to vehicle with potentially less tumor infiltration. These findings highlight the potential of carboplatin-gemcitabine as a promising chemotherapeutic combination for RB patients.

Retinoblastoma is a challenging disease. I congratulate the group on their efforts to help push the knowledge of RB treatment forward. I think this study has merit but some experimental and statistical issues currently prevent a recommendation for publication without addressing the following points:

Major points:

1. The two cell lines were grown in different conditions (20% vs 10% FBS, and RPMI vs DMEM). It is unclear how much of the discrepancy in response is due to this. One recommended experiment is to condition both lines to the opposite condition to validate the hits in a 2x2 matrix. That these are suspension/spheroidical vs adherent cells should also be made clear and discussed as potential confounder.

2. RPE-1 are hTERT-immortalized cells. This wasn’t made clear in the paper and can have implications in the interpretation of the study as this line was used as reference to which Y79 data were compared. Importantly, describing such a line as “normal” is problematic; “non-cancerous” or “non-transformed” would be more appropriate.

3. The study bases all conclusions on only the Y79 line in both in vitro and in vivo studies. The authors should discuss this limitation in the paper.

4. Another HTS synergistic screen was conducted using Y79 cells previously by Mahida et al. (https://doi.org/10.1371/journal.pone.0059156). I could not find a mention of this work in this study. How does this prior study compare with the current study? were there consistent and inconsistent observations?

5. There seems to be no data presented on drug treatment and thermotherapy in vivo. Given the premise of the manuscript, how do these drugs or their combination perform as a chemothermotherapy in vivo?

6. Bar graphs in Fig2D, 2E, 3B, and 3D are difficult to read as the dots were made the same color as the bars, so it is difficult to see the distribution. A box plot may be more appropriate and easier to visualize here or change the dots into circles of a different color (e.g. black).

7. Most of the above mentioned bar graphs do not show p-values or statistical comparisons, making the figure difficult to interpret.

8. The study should evaluate the statistical significance of synergy in Fig2 analyses (It appears the paper is using SynergyFinder 1.0, but the p-value calculations are likely offered in newer versions: https://www.bioconductor.org/packages/release/bioc/vignettes/synergyfinder/inst/doc/User_tutorual_of_the_SynergyFinder_plus.html).

9. Line 422: This effect is not statistically significant from Fig 3D and should be stated clearly that there was no effect of the combination on EdU incorporation. Moreover, it is odd that the p-value is > 0.1 between DMSO and gemcitabine alone in Fig 3D. It appears this is an effect of only one outlier in the DMSO group. The location of the dots in gemcitabine also appear higher than what the bar graph is indicating (the dots are difficult to see). Taken together, conclusions based on the current results are problematic and require additional replicates given the variability. It’s also unclear what each dot represents (different fields of view in the same well? different wells? different experiment dates?).

10. In Fig 4, the optic nerve appears to be double the number of animals used likely because two optic nerves are being counted as independent experiments/samples from the same animal. This breaks the assumption of the Fisher exact test that observations must be independent. One solution would be to count the number of animals with any positive tumor infiltration in the optic nerve, or increase the number of animals which will allow a better study of brain infiltration as well.

11. In Fig 4, how does carboplatin alone compare to gemcitabine or the combination? This is a key control.

12. Line 528-531: The statement is somewhat misleading. The in vivo data shows that there was no statistical difference in survival between standard of care melphalan and/or gemcitabine with the combination of gemcitabine+carboplatin. The authors can make this clearer by breaking the discussion on survival and tumor infiltration into separate statements.

Minor edits:

1. Line 221: Spelling: cone-rod homeobox RNA

2. Figure 1: Spelling: Under ‘Second screened out’ box, the word ‘Highly’ is misspelled.

3. Figure 4: Clarity: The colors of the lines are very similar to one another. Perhaps different colors or dashed lines could be used instead.

Reviewer #4: Manuscript Number: PONE-D-24-22008

PLOS ONE

Research Article

“High-throughput screening of drug libraries identifies a new synergistic drug combination for the treatment of retinoblastoma”

The authors identify gemcitabine, a nucleoside analogue inhibitor of DNA synthesis, as a potential re-purposed retinoblastoma (RB) therapy when applied in synergistic combinations with thermotherapy and/or cytotoxic agent carboplatin. Gemcitabine was identified through HTS on a set of FDA-approved compounds and some 80 additional anti-cancer agents. Compounds were tested for efficacy on target RB (Y79 cell line) and toxicity in RPE-1 (normal retinal pigment epithelial) as a counterscreen. Follow-up mechanistic assays and efficacy testing in orthotopic xenograft mice models of RB add high value to this work. The paper is well-written, experimental design and reporting is sound, and figures/results are clear and effective. Great work! I recommend publication with no revisions.

Reviewer #5: - Fig 3A & B: It would be better if authors add the data for the negative control/ vehicle control used in these experiments

- Fig 3C & D: To be clear on the whole experiment, it is advised to add the carboplatin alone data to figure 3C and 3D

- It is advised to discuss about potential toxicity studies to evaluate any side effects and overall safety of the gemcitabine and carboplatin combination, particularly focusing on pediatric models to ensure relevance to the target patient population.

- While the potential benefits of combining carboplatin and gemcitabine with thermotherapy were suggested, detailed experimental data on the efficacy and safety of this combined approach in vivo were not provided. Are there any in vivo experimental models authors would like to test their suggested treatment options (chemotherapy+ thermotherapy) or would like to include in the discussion as a future direction of this study?

**Do you want your identity to be public for this peer review?** For information about this choice, including consent withdrawal, please see our Privacy Policy

Reviewer #1: No

Reviewer #2: **Yes:** Markus de Raad

Reviewer #3: No

Reviewer #4: No

Reviewer #5: No

---

## [Author Response · Author response to Decision Letter 1]

28 May 2025

Authors response to Reviewers comments

Reviewer #1:

1. Page 12, line 267 – For the 21 drugs advanced that showed enhancement with thermotherapy, why did the authors choose only drugs that were not active at 37oC? Wouldn’t it be more advantageous to have a drug that both works at normal body temperature and then is further enhanced with thermotherapy?

We thank the reviewer for raising this question. The main clinical challenge we address in this work is whether it is possible to minimize side effects associated with chemotherapy. Chemotherapeutics are administered systemically in the chemothermotherapy clinical protocol, and the amount of drug distributed to the eye is considerably low due to blood-retinal barrier limited crossing. Since retinoblastoma patients are typically very young children and higher dose of systemic chemotherapy results in serious associated toxicity, we were looking for drugs that are inactive at body temperature and can be selectively activated at the tumor site.

The following text has been added to the manuscript to clarify our intent:

“We then aimed to select drugs that are inactive at body temperature but become active at elevated temperatures to mitigate the clinical challenge of chemotherapy-associated toxicity.”

2. Page 15, line 326 – I would recommend adding the avg values for carboplatin-etoposide and carboplatin-vincristine in the text as well.

We thank the reviewer for this comment – the requested changes have been made.

3. Page 15, Line 341 – authors state that the combination of carboplatin (3.3 uM) and gemcitabine (2.5 uM) only resulted in 60% cell inhibition. Yet, the IC50 of gemcitabine alone is listed as 0.045 uM. Similarly, the dose-response curve in Figure 1 for gemcitabine shows ~60% inhibition at concentrations of 0.1 uM. If the curve in Figure 1 is to be believed, gemcitabine at 2500 nM (Log ~3.4) would be expected to be close to 100% inhibition on its own. Why the big discrepancy in cell viability/% inhibition values between Figure 1 and Figure 2?

We appreciate the reviewer’s careful analysis and valid observation regarding the discrepancy in cytotoxicity values between Figures 1 and 2. This discrepancy can be attributed to several factors inherent to cell-based cytotoxicity assays, such as cell batch variation, biological and technical variability, and edge effects. The following sentence was added to address this point in the manuscript:

“Discrepancies in absolute cytotoxicity values across experiments likely reflect inherent inter-assay variability, including differences in cell passage, plating uniformity, and positional effects. Importantly, relative response patterns remained internally consistent.”

4. Page 15, Line 342 – Quantitating synergistic effects is not trivial to perform and even more difficult to describe. For example, the line “This corresponds to a 15-fold reduction in the concentration of carboplatin required to produce such an effect when compared to carboplatin alone”, yet according to Figure 2, this same % inhibition (~60%) is observed even when 0 uM of carboplatin is used. It is also unclear where the 15-fold reduction number comes from as 15 fold of 3.3 uM is ~50uM and yet 50uM in Figure 2a with no Gemcitabine only inhibited ~11%.

The reviewer is correct in noting that the statement regarding a “15-fold reduction in the concentration of carboplatin required to produce such an effect” was based solely on a comparison to carboplatin monotherapy (IC50 value ~50 µM) and did not account for the effects of gemcitabine alone. As our primary objective in this section was to explore the potential of carboplatin-gemcitabine combinations as improvements over standard therapies, we focused our comparison on carboplatin, which is commonly used in the clinical setting.

However, we acknowledge that without incorporating the effects of gemcitabine alone into the analysis, the claim regarding fold-reduction is incomplete and potentially misleading. In light of this, we have removed this sentence in question from the manuscript.

5. Page 16, line 368 – Authors state “At these doses, carboplatin-gemcitabine resulted in 60% cell viability inhibition in Y79 cells compared to 20% in RPE-1 cells, indicating a promising therapeutic window.” However, the numbers they quote do not appear to be what is depicted in Figure 2c, carboplatin-gemcitabine in Y79 cells looks to be ~60% inhibition as they state, but in RPE-1 cells, the inhibition looks to be even higher inhibition ~65%. No ideas where authors get 20% cell inhibition value from. The figure in 2c makes it appear that there is no therapeutic window using any of the chemotherapeutics as almost all Cell inhibition is similar between Y79 and RPE-1 cells.

Similarly, the other cell inhibition percentages for the other drugs do not appear to be correct either. Authors state 25% and 18% inhibition for carboplatin-irinotecan in Y79 and RPE-1 cells, respectively. However, the graph in Figure 2c clearly shows a greater amount of cell inhibition for the RPE-1 cells.

We appreciate the reviewer’s careful attention to the data presented in Fig 2c which does not align with the cell viability values reported in the text. This was due to an error in the original graph, which inadvertently displayed incorrect data.

We have corrected Fig 2c to accurately reflect the experimental results. The updated figure now corresponds to the text values, including the differential effects observed between Y79 and RPE-1 cells. We apologize for the oversight and have carefully reviewed all other figures and data points to ensure consistency and accuracy throughout the manuscript.

6. Page 18, line 397 – Authors state “Furthermore, drug combinations induced greater levels of apoptosis than single drugs as determined by flow cytometry (Fig. 3a,b).” Yet, the flow graphs in Figure 3a do not back up this statement. In fact, Gemcitabine alone had 54.2% in the early apoptotic region and 19.2% in the late apoptotic/dead cell region. Yet the carboplatin + gemcitabine only had 46.9% in early and 12.4% in late. Similarly, etoposide alone appeared to do better in apoptotic induction as compared to the carboplatin + etoposide.

For the quantitation in Figure 2b, were the graphs in figure 2a not used for the analysis? I assume the dots on the bar chart are the individual data points. So if we take etoposide as an example, why is there no dot at 66% (or 76% if you are combining early and late apoptosis regions)?

We thank the reviewer for their careful assessment and for pointing out the discrepancy between the quantitative claim and the data shown in Figure 3a (now, S6 Fig). Upon reevaluation, we agree that the original flow cytometry plots do not clearly support the statement that drug combinations induced higher levels of apoptosis than single agents in all cases.

To address this issue, we have revised the manuscript text for accuracy and clarity. Specifically, the sentence previously stating that "drug combinations induced greater levels of apoptosis than single drugs" has been modified to a more descriptive and data-driven summary of the Annexin V staining results, as follows:

“Furthermore, induction of apoptosis by the drug combinations was assessed using Annexin V staining (S6 Fig). In the samples treated with either carboplatin, gemcitabine or etoposide alone, the fraction of early apoptotic cells was 26, 54 and 49%, respectively. The number of early apoptotic cells in the samples treated with either carboplatin–gemcitabine (66%) or carboplatin–etoposide (59%) was higher than with either of the single treatments. However, the proportion of late apoptotic cells was lower in the combinations.”

This revised statement more accurately reflects the observed data, distinguishing between early and late apoptotic populations rather than combining them into a general “apoptotic” category.

Regarding Fig 3b, we acknowledge the reviewer’s valid concern about the traceability of the data points to the flow cytometry plots in Fig 3a. To resolve this issue and avoid potential confusion, we have removed the quantification bar graph (Fig 3b) from the revised manuscript and retained only the representative flow cytometry plots in Fig 3a. This change ensures that all presented data are directly supported by the shown experimental results.

We appreciate the reviewer’s attention to detail, which helped improve the clarity and accuracy of the manuscript.

Reviewer #2:

This manuscript presents the high-throughput screening of a drug library to identify candidates for synergistic drug combinations for the treatment of retinoblastoma. The study is straightforward and clearly described and the manuscript is well written. However, I believe there are some (minor) issues that need to be addressed by the authors.

1. Aim of library screening

In the introduction, the authors clearly describe the goal of the high-throughput screening, to identify candidates that are both active for RB (cytotoxic to Y79 cells) and selective (non-toxic to RPE-1) with and without thermotherapy. However, after identifying candidates through the high-throughput screen, candidates are tested in drug combinations and the focus shifts towards multi-drug treatments and away from thermotherapy. If the goal is to identify candidates for multi-drug treatments, why not include combinations with carboplatin in the initial screen? And why use the delta HTS parameter for score if the downstream combination with carboplatin are performed at 37C? Also, authors introduce the drug library as a general drug library of approved drugs. But during hit selection, pharmacologic targets (line 305) are used as a criterion. So, why not only screen drugs with relevant pharmacologic targets? Or why exclude hits with no relevant target but identified through the screen? And it was little disappointing to found out at the end of the discussion that gemcitabine and platinum-based drug combinations are already used as effective treatments. The authors should address these issues and explain in more detail why certain choices were made. Furthermore, the authors should explain what the third round of screening entailed and why it was performed. And the scheme

We thank the reviewer for raising an important question regarding our research strategy. Our ultimate goal is to mitigate current clinical challenge of minimizing chemotherapy-related cytotoxicity in retinoblastoma treatment. This study is an attempt to systematically evaluate existing drugs in close relation to current clinical treatment strategies. Hence, we started with high-throughput screening of single drugs, then explored their synergy with each other. To ensure the feasibility of further studies, we assumed that drugs that have a relevant target will be more likely to get accelerated approval, since pediatric oncology is a reasonable conservative field. These selected pairs were then investigated with and without thermotherapy in matrix experiments and corresponding deltaHTS parameter values are presented in S3 Fig. We were encouraged by discovering that gemcitabine-carboplatin combination is already employed in the clinic for adult patients, as this supports its efficacy and toxicity in the clinic. The following changes were made to clarify the intentions of this study:

Introduction, line 95: “Our overarching goal was to identify candidate drugs that could enhance or complement current chemotherapy regimens, either as monotherapies or in rationally designed combinations, with or without thermotherapy.”

Results, line 329: “To build on the findings from the single-agent screen and explore clinically actionable regimens, we next investigated rational drug combinations with carboplatin, a part of multi-drug regimens in RB treatment(31).”

Results, line 362: “Among the evaluated drug combinations, carboplatin-gemcitabine exhibited an increase in average synergy score (𝛿avg) from 18.75 at 37 °C to 27.53 at 42 °C, although this trend did not reach statistical significance. Consequently, subsequent investigations focused on evaluating the efficacy of the drug combinations in retinoblastoma models independently of thermotherapy.”

2. Minor issues

- Line 170-171: Sentence repeated

We thank the reviewer for this comment; the repeated sentence was deleted.

3. - Line 237: Approved by who? Add to text

We thank the reviewer for this comment; the sentence was clarified:

“The library consists of the Prestwick Chemical Library and selected Enzo Kinase Inhibitor library made up of 1360 off-patent, predominantly FDA-approved & EMA-approved drugs.”

4. - Line 241-242: Please explain the scoring the main text. It is explained in the methods section, but this is important for the results section.

We thank the reviewer for this comment; the following text was added to improve clarity of the experimental workflow:

“HTS scores were calculated by normalizing results to controls for each plate, where a score of 0 represents low cytotoxicity of the negative control, and a score of 1 corresponds to high cytotoxicity of the positive control. Hit compounds were identified when their scores surpassed the average of negative controls plus three times the standard deviation (3•SD).”

5. - Line 325: Are the authors comparing the observed in vitro screening results with other in vitro results? If so, please add reference. If there refer to actual clinical data, the authors should not compare in vitro screening data to clinical treatments.

In this study, we compared the results of drug synergy experiments conducted in our work, so no reference needs to be added.

6. - Line 533-549: A lot of speculation. The authors should tone down this section.

We thank the reviewer for pointing out to this section. We have toned down the section and the revised text is copied below:

“To provide a preliminary reference point for potential clinical translation, we compared the intravitreal dose of gemcitabine used in our preclinical model to that of melphalan, the current standard for intravitreal chemotherapy in retinoblastoma, which is typically administered at doses of 20–30 µg(69). In the mouse experiments, 0.05 ng of gemcitabine was injected into an eye with a vitreous volume of approximately 5 µL. Assuming a vitreous volume of 3.4 mL for a 5-year-old child, based on ocular growth data(70,71), this would be expected scale to an estimated human-equivalent dose of ~34 ng. While this extrapolation is based solely on proportional volume scaling and does not account for interspecies differences in drug distribution, clearance, or retinal penetration, it suggests that gemcitabine may lower intravitreal doses compared to current agents.

Given that standard intravitreal therapies have shown high efficacy in managing vitreous seeds, we do not propose gemcitabine as a replacement in frontline treatment. Instead, its role may potentially lie in refractory intraocular retinoblastoma, where conventional options are limited, or in metastatic disease through systemic or intra-arterial routes. Notably, both gemcitabine and its combination with platinum-based drugs have been used clinically in pediatric oncology(72,73) , which may support further investigation of this approach. Nonetheless, these findings are exploratory and require substantial additional validation, including safety profiling and pharmacokinetic studies, before consideration for clinical application.”

7. - I'm missing a conclusion section or conclusion paragraph at the end of the manuscript

The following concluding paragraph was included at the end of Discussion section:

“In conclusion, this study aimed to address the clinical challenge of limited treatment options for retinoblastoma by employing a systematic approach to identify effective therapeutic strategies. Through high-throughput screening of 1,360 compounds, we evaluated single agents, drug combinations, and the integration of chemotherapy with thermotherapy, modalities widely used in retinoblastoma treatment. Our findings highlight the gemcitabine-carboplatin combination as a promising candidate, demonstrating efficacy comparable to standard-of-care treatments in preclinical models. While further investigation is required to assess its clinical applicability, this study provides a foundation for expanding the therapeutic landscape

---

## [Decision Letter · Decision Letter 1]

10 Jul 2025

Dear Dr. Dyson,

Thank you for submitting your manuscript to PLOS ONE. After careful consideration, we feel that it has merit but does not fully meet PLOS ONE’s publication criteria as it currently stands. Therefore, we invite you to submit a revised version of the manuscript that addresses the points raised during the review process.

Following the review of your manuscript titled "High-throughput screening of drug libraries identifies a new synergistic drug combination for the treatment of retinoblastoma" I recommend that it should be revised taking into account the changes requested by the reviewer. Since the requested changes are minor, the revised manuscript will undergo a second round of review by the same reviewer.

We look forward to receiving your revised manuscript.

Kind regards,

Mounir Tilaoui, Ph.D

Academic Editor

PLOS ONE

Journal Requirements:

Reviewers' comments:

Reviewer's Responses to Questions

**Comments to the Author**

Reviewer #2: (No Response)

Reviewer #5: All comments have been addressed

2. Is the manuscript technically sound, and do the data support the conclusions?

Reviewer #2: Yes

Reviewer #5: Yes

3. Has the statistical analysis been performed appropriately and rigorously?

Reviewer #2: I Don't Know

Reviewer #5: Yes

4. Have the authors made all data underlying the findings in their manuscript fully available?

Reviewer #2: No

Reviewer #5: Yes

5. Is the manuscript presented in an intelligible fashion and written in standard English?

Reviewer #2: Yes

Reviewer #5: Yes

Reviewer #2: I would like to thank the authors for the responses to my questions and concerns. The authors made substantial changes to the manucript which improved the overall quality of the presented work.

However, there are still some minor issues that need to be addressed.

Line 279-281: "Out of the 38 identified drugs, the 11 which scored highest in the Y79 cell line in a third round of screening, with an HTS score 280 of over 0.9, were chosen for further investigation". The authors need to include the data for this third round of screening and should describe this third round of screening in the manuscript. Since the selections were based on the all 3 rounds, the data, results and outcomes of this third round of screening should be presented as findings in the manuscript. The given statement is not enough.

Reviewer #5: (No Response)

**Do you want your identity to be public for this peer review?** For information about this choice, including consent withdrawal, please see our Privacy Policy

Reviewer #2: **Yes:** Markus de Raad

Reviewer #5: No

---

## [Author Response · Author response to Decision Letter 2]

28 Jul 2025

Authors response to Reviewers comments

Reviewer #2:

Line 279-281: "Out of the 38 identified drugs, the 11 which scored highest in the Y79 cell line in a third round of screening, with an HTS score 280 of over 0.9, were chosen for further investigation". The authors need to include the data for this third round of screening and should describe this third round of screening in the manuscript. Since the selections were based on the all 3 rounds, the data, results and outcomes of this third round of screening should be presented as findings in the manuscript. The given statement is not enough.

We thank the reviewer for this comment. We recognize that the phrase "third round of screening" may have led to confusion. To clarify, this step did not involve a separate experimental screen but rather a data-driven prioritization of the 38 hits identified active to Y79 cells in the first-pass screen and inactive to RPE-1 cells in the counter screen. Specifically, we selected compounds with the highest HTS scores (HTS > 0.9) for further investigation, based on their strong cytotoxic effect in Y79 cells. This stringent threshold was used to identify the most potent compounds; under the rationale that highly cytotoxic drugs may achieve therapeutic efficacy at lower doses. This is particularly relevant in pediatric oncology, such as retinoblastoma, where minimizing systemic toxicity and reducing drug burden are key clinical considerations. Furthermore, to enhance clarity, we revised Fig.1a and introduced the following naming for each step of primary hits identification. Thus, “first-pass screen”, experimental, Y79 37°C; “counter screen”, experimental, RPE-1 37°C; “strong hits”, data-driven prioritization; “hyperthermia screen”, experimental, Y79 42°C; “thermosensitive hits”, data-driven prioritization; “heat-only hits”, data-driven prioritization.

The data supporting these selection process is in S1 Table (65 compounds from the “first-pass screen”, 11 “strong hits” highlighted) and in S2 Table (68 compounds from the “hyperthermia screen”, 21 “heat-only hits” highlighted).

The following changes have been made in the manuscript to reflect this clarification:

“To further prioritize compounds with strong anti-cancer potential, we applied a data-based selection step by ranking these 38 compounds by their HTS scores in Y79 cells and selecting those with an HTS score > 0.9. This resulted in 11 compounds with the highest cytotoxic activity at 37 °C, which were advanced for further investigation (Fig. 1a)."

---

## [Decision Letter · Decision Letter 2]

7 Oct 2025

Dear Dr. Dyson,

Thank you for submitting your manuscript to PLOS ONE. After careful consideration, we feel that it has merit but does not fully meet PLOS ONE’s publication criteria as it currently stands. The manuscript had been significantly improved after two revisions, however there are still a few minor issues that have to be addressed. Therefore, we invite you to submit a revised version of the manuscript that addresses the points raised during the review process.

We look forward to receiving your revised manuscript.

Kind regards,

Irina V. Lebedeva, Ph.D.

Academic Editor

PLOS ONE

Journal Requirements:

Reviewers' comments:

Reviewer's Responses to Questions

**Comments to the Author**

Reviewer #2: All comments have been addressed

Reviewer #6: (No Response)

Reviewer #7: (No Response)

Reviewer #8: (No Response)

2. Is the manuscript technically sound, and do the data support the conclusions?

Reviewer #2: Yes

Reviewer #6: Yes

Reviewer #7: Yes

Reviewer #8: Yes

3. Has the statistical analysis been performed appropriately and rigorously?

Reviewer #2: Yes

Reviewer #6: Yes

Reviewer #7: Yes

Reviewer #8: Yes

4. Have the authors made all data underlying the findings in their manuscript fully available?

Reviewer #2: Yes

Reviewer #6: Yes

Reviewer #7: Yes

Reviewer #8: Yes

5. Is the manuscript presented in an intelligible fashion and written in standard English?

Reviewer #2: Yes

Reviewer #6: Yes

Reviewer #7: Yes

Reviewer #8: Yes

Reviewer #2: (No Response)

Reviewer #6: Thank you for the opportunity to review your article on drug discovery in retinoblastoma and the candidacy of gemcitabine-carboplatin. I find that the methods and statistical analysis are strong and adequate to test your hypothesis. At this time I believe that there remains some revisions before this article is acceptable for publication. Please see my comments below.

Intro:

- 81-82) “carboplatin with etoposide with or without vincristine” what are the routes of administration (ROA) of these medications? Unclear if standard of care is systemic or intravitreal based on text.

- 90-101) no indication of ROA being intravitreal. Please make more clear what the in vivo experiments are testing and make sure it is reflected elsewhere in the article including abstract.

Methods:

- How was eye survival determined? Difficult to ascertain significance of this metric without this defined.

- This was addressed in discussion, but why only use one RB cell line for identification of active compounds in vitro? Does this not restrict generalizability?

- 135) 42 °C in the first hour and then transferred to incubator at 37 °C. — this is enough to mimic thermotherapy? Citation for this proof of concept?

Results:

362) carboplatin-vincristine ( avg of -2.98) clinically used but < 0 relates to an antagonistic effect and 417) carboplatin-vincristine demonstrated a strong antagonistic effect ( < -10).

- This was briefly explored in discussion but please elaborate. Does this finding contrast to prior literature? What implication does this finding have on your results if any? What are some explanations behind this?

Discussion:

633-635) grade 3/4 neutropenia and thrombocytopenia “are generally manageable”. “Manageable” does not seem to capture the gravity of this type of adverse event. Consider rephrasing and including a clinical citation.

Thank you again for the chance to review this compelling manuscript!

Reviewer #7: The article describes the process and results of screening a large number of chemotherapeutic agents to identify a drug that is at the same time cytotoxic to the tumour and gentle on RPE cells. I also find from the previous comments and clarifications given that the authors deliberately aimed at finding drugs that are inactive at normal temperature and become active only on thermotherapy. As a clinician I have the following comments to make.

1. In thermo-chemotherapy, thermotherapy is an addendum to the chemotherapy. The ability of the laser that is used to perform TTT (transpupillary thermotherapy) (to penetrate the tumour depth is limited, and hence large tumours cannot be heated with TTT throughout the extent of the tumour while chemotherapy can circulate to the entire extent of the tumour. Hence it is important that a chemo therapeutic agent should be active at body temperature as well as at higher temperature attained by thermotherapy.

In this respect the authors have been self-contradictory in their replies to previous reviewers.

Eg. The following is the explanatory note for reviewer 1 comment 1. ‘Since retinoblastoma patients are typically very young children and higher dose of systemic chemotherapy results in serious associated toxicity, we were looking for drugs that are inactive at body temperature and can be selectively activated at the tumor site.’ – Seems to clearly state that the intent of this experiment is to identify only drugs that are inactive at body temperature.

However, in the same version while replying to comments of reviewer 2, comment 1- the authors state ‘“Our overarching goal was to identify candidate drugs that could enhance or complement current chemotherapy regimens, either as monotherapies or in rationally designed combinations, with or without thermotherapy.”

Hence the stress by the authors should be only of identifying potential new chemotherapeutic drugs for retinoblastoma with or without added efficacy enhancement with thermotherapy.

Otherwise I have no specific comments on methodology, results and interpretation which have been extensively reviewed and corrected already.

Reviewer #8: This manuscript by Tseng et al. presents an HTS and validation of compounds synergistic for selective retinoblastoma toxicity. This is a revised manuscript, but this reviewer is seeing it for the first time. Overall it a really well written and well conceived manuscript. There are some minor technical issues that do not take away from the overall conclusions drawn from this work, but addressing these will further aid a reader’s overall understanding and reproducibility of the work described here.

1. how were the concentration ranges chosen for dose response assays (lines 158-163)?

2. does seeding RPE-1 cells with compounds already in the growth media affect attachment/morphology (line 138)?

3. was the switch from a 72 hr viability assay to a 5 day with the combination therapy necessitated by the switch from 384 to 96 well?

4. what was the seeding density in the 96 well assay (line 176), and why did your dosing strategy change with the combination from initial screening (line 178 vs 136)?

5. how were the Y79 cells growing that they could be stained and imaged in a 96 well plate (lines 200-208)? They can grow semi-attached but usually don’t stay that way through washes. It might be helpful to include arrows pointing to the EdU incorporated cells.

6. Does the lack of inhibition by clinically used carboplatin-etoposide combination seen in this synergy experiment point to any issues with the other conclusions drawn from this in vitro work? (line 414)

7. Figure legends: please include statistical test, n, and indications of what asterisks mean

8. Figure 3 title is an overstatement – the EdU analysis does not show “mechanism of action”. Please rephrase.

9. The authors use a variable slope model to analyse their dose response curves. How many data points did they have between 0.78 to 100uM (how many are needed to use variable slope vs standard slope?)

10. Please mention the post hoc test used for one way ANOVA (line 250)

11. It would be helpful if it were possible to include tumor/eye images for the in vivo study

**Do you want your identity to be public for this peer review?** For information about this choice, including consent withdrawal, please see our Privacy Policy

Reviewer #2: **Yes:** Markus de Raad

Reviewer #6: No

Reviewer #7: No

Reviewer #8: No

---

## [Author Response · Author response to Decision Letter 3]

19 Nov 2025

Authors response to Reviewers comments

Reviewers' comments:

Reviewer #6:

Thank you for the opportunity to review your article on drug discovery in retinoblastoma and the candidacy of gemcitabine-carboplatin. I find that the methods and statistical analysis are strong and adequate to test your hypothesis. At this time I believe that there remains some revisions before this article is acceptable for publication. Please see my comments below.

Intro:

- 81-82) “carboplatin with etoposide with or without vincristine” what are the routes of administration (ROA) of these medications? Unclear if standard of care is systemic or intravitreal based on text.

We appreciate the reviewer’s attention to treatment details. The sentence in question aims to illustrate commonly used chemotherapeutic agents and their combinations rather than to summarize specific treatment protocols. Carboplatin with etoposide, with or without vincristine, is administered systemically. This detail has been added to the sentence (line 81).

- 90-101) no indication of ROA being intravitreal. Please make more clear what the in vivo experiments are testing and make sure it is reflected elsewhere in the article including abstract.

We thank the reviewer for this observation. The following changes were made:

Lines 98-102: “Furthermore, the mechanism of action of a selected candidate was elucidated in vitro, and its efficacy and safety were evaluated in vivo following intravitreal administration and compared to current standard-of-care treatments (intravitreal melphalan or systemic combination of etoposide and carboplatin).”

Methods:

- How was eye survival determined? Difficult to ascertain significance of this metric without this defined.

We thank the reviewer for this comment. Eye survival was defined as the time from treatment initiation to the pre-specified humane endpoint, when the affected eye reached three times its normal size, at which point enucleation was performed. This definition is now explicitly referenced in the “Efficacy studies of intravitreal gemcitabine” section and cross-linked in the “Statistical analysis” paragraph to improve clarity.

- This was addressed in discussion, but why only use one RB cell line for identification of active compounds in vitro? Does this not restrict generalizability?

We thank the reviewer for this comment. We selected Y79 for the primary screen to maximize assay robustness and to impose stringent selection pressure. Y79 models a more aggressive retinoblastoma phenotype, so compounds advancing from this screen are unlikely to be false positives under milder conditions. In addition, only two widely available RB lines exist (Y79 and WERI-Rb1), which constrains broad parallel screening within the current scope. We acknowledge that generalizability would benefit from testing in additional models and have noted this as a limitation.

- 135) 42 °C in the first hour and then transferred to incubator at 37 °C. — this is enough to mimic thermotherapy? Citation for this proof of concept?

We thank the reviewer for this comment. Appropriate citations were added to the Results section (line 296).

Results:

- 362) carboplatin-vincristine ( avg of -2.98) clinically used but < 0 relates to an antagonistic effect and 417) carboplatin-vincristine demonstrated a strong antagonistic effect ( < -10).

This was briefly explored in discussion but please elaborate. Does this finding contrast to prior literature? What implication does this finding have on your results if any? What are some explanations behind this?

We thank the reviewer for this comment. In our Y79 model, carboplatin–vincristine showed antagonism within the tested concentration range. To our knowledge, no combination-synergy studies have been reported in retinoblastoma. Reports from other tumor types are inconsistent (some show no synergistic effect, others moderate synergy) indicating that outcomes are highly cell-line and dose dependent(1–3). The observed antagonism in Y79 therefore reflects model-specific interactions.

Discussion:

- 633-635) grade 3/4 neutropenia and thrombocytopenia “are generally manageable”. “Manageable” does not seem to capture the gravity of this type of adverse event. Consider rephrasing and including a clinical citation.

We thank the reviewer for this comment. The following changes were made:

Line 647-649: “In retinoblastoma, severe hematologic events are documented with both intra-arterial and systemic chemotherapy and require close clinical monitoring and, when indicated, dose modification(85,86).”

Thank you again for the chance to review this compelling manuscript!

Reviewer #7:

The article describes the process and results of screening a large number of chemotherapeutic agents to identify a drug that is at the same time cytotoxic to the tumour and gentle on RPE cells. I also find from the previous comments and clarifications given that the authors deliberately aimed at finding drugs that are inactive at normal temperature and become active only on thermotherapy. As a clinician I have the following comments to make.

1. In thermo-chemotherapy, thermotherapy is an addendum to the chemotherapy. The ability of the laser that is used to perform TTT (transpupillary thermotherapy) (to penetrate the tumour depth is limited, and hence large tumours cannot be heated with TTT throughout the extent of the tumour while chemotherapy can circulate to the entire extent of the tumour. Hence it is important that a chemo therapeutic agent should be active at body temperature as well as at higher temperature attained by thermotherapy.

In this respect the authors have been self-contradictory in their replies to previous reviewers.

Eg. The following is the explanatory note for reviewer 1 comment 1. ‘Since retinoblastoma patients are typically very young children and higher dose of systemic chemotherapy results in serious associated toxicity, we were looking for drugs that are inactive at body temperature and can be selectively activated at the tumor site.’ – Seems to clearly state that the intent of this experiment is to identify only drugs that are inactive at body temperature.

However, in the same version while replying to comments of reviewer 2, comment 1- the authors state ‘“Our overarching goal was to identify candidate drugs that could enhance or complement current chemotherapy regimens, either as monotherapies or in rationally designed combinations, with or without thermotherapy.”

Hence the stress by the authors should be only of identifying potential new chemotherapeutic drugs for retinoblastoma with or without added efficacy enhancement with thermotherapy.

Otherwise I have no specific comments on methodology, results and interpretation which have been extensively reviewed and corrected already.

We thank the reviewer for this comment. Our study design included two coordinated objectives: first, to identify drugs with intrinsic activity against retinoblastoma at physiological temperature, and second, to explore whether mild hyperthermia could further enhance their cytotoxicity. The sentence in question was phrased too narrowly and may have implied that only heat-activated compounds were sought. In fact, both thermally enhanced and baseline-active compounds were retained for downstream analysis, as reflected in the Results and Supplementary Tables. We clarified this point in the text to prevent misinterpretation.

Line 308-311: “This set includes agents with strong baseline activity at physiological temperature as well as agents whose cytotoxicity is specifically enhanced by mild hyperthermia, indicating complementary routes for potential chemotherapeutic and chemothermotherapeutic development.”

Reviewer #8:

This manuscript by Tseng et al. presents an HTS and validation of compounds synergistic for selective retinoblastoma toxicity. This is a revised manuscript, but this reviewer is seeing it for the first time. Overall it a really well written and well conceived manuscript. There are some minor technical issues that do not take away from the overall conclusions drawn from this work, but addressing these will further aid a reader’s overall understanding and reproducibility of the work described here.

1. how were the concentration ranges chosen for dose response assays (lines 158-163)?

We thank the reviewer for this question. Published IC₅₀ values for standard retinoblastoma chemotherapeutics were reviewed to contextualize the concentration ranges tested in this study and to ensure they covered a pharmacologically meaningful window for retinoblastoma cells. The following changes were made:

Line 158-161: “To assess dose-dependent drug responses of the identified hits, Y79 cells were plated at a density of 4.0•105 cells per well, in 384-well plates, and exposed to each compound in the concentration range 0.78 to 100 µM (the range was chosen based on available Rb cytotoxicity data(22–25).”

2. does seeding RPE-1 cells with compounds already in the growth media affect attachment/morphology (line 138)?

We thank the reviewer for this question. We did not specifically evaluate the effect of compound pre-dispensing on RPE-1 attachment. The reverse assay format, in which compounds were plated first, and cells were added afterward, was selected based on technical feasibility and consistency with our screening setup. Plate-level controls did not indicate attachment-related artifacts.

3. was the switch from a 72 hr viability assay to a 5 day with the combination therapy necessitated by the switch from 384 to 96 well?

We thank the reviewer for this question. The 5-day readout in the combination assay was chosen to resolve synergy at low, clinically relevant concentrations and to capture slower combination effects that were less distinct at 72 h. Primary screening, dose-response studies, and combination matrices used 384-well plates at 72 h for throughput; low concentration combination matrices were run in 96-well plates for practical reasons (matrix layout and volumes), but the 5-day endpoint was a design choice to improve discrimination, not a requirement of the 96-well format.

4. what was the seeding density in the 96 well assay (line 176), and why did your dosing strategy change with the combination from initial screening (line 178 vs 136)?

We thank the reviewer for this question. The seeding density remained consistent throughout the experiments and was Y79 (4.0•105 cells/mL) and RPE-1 (1.0•105 cells/mL). The following changes were made in the methods section:

Line 132-134: “30 µl of cell suspension was added for the Y79 (4.0•105 cells/mL) and RPE-1 (1.0•105 cells/mL).”

Line 158-162: “To assess dose-dependent drug responses of the identified hits, Y79 cells were plated at a density of 4.0•105 cells/mL, in 384-well plates, and exposed to each compound in the concentration range 0.78 to 100 µM (the range was chosen based on available Rb cytotoxicity data(22-25).”

Line 178-180: “Y79 cells were plated at a density of 4.0•105 cells/mL into 96-well plates, and the following day, treated with serial dilutions of carboplatin, gemcitabine, or their combination.”

5. how were the Y79 cells growing that they could be stained and imaged in a 96 well plate (lines 200-208)? They can grow semi-attached but usually don’t stay that way through washes. It might be helpful to include arrows pointing to the EdU incorporated cells.

We thank the reviewer for this observation. Y79 cells were seeded and treated directly in 96-well plates and subsequently transferred to microcentrifuge tubes for the EdU assay. After staining, the cell suspensions were gently transferred to glass-bottom imaging dishes for microscopy. This workflow avoided attachment-related issues typical for Y79 and ensured minimal cell loss during washing. The following details were added:

Line 206-208: “After being fixed with 4% paraformaldehyde for 30 min, cells were transferred to microcentrifuge tubes, treated with 0.1% Triton X-100 for 20 min and washed with phosphate-buffered saline three times.”

Line 210-211: “Cells were gently transferred to glass-bottom imaging dishes for microscopy.”

6. Does the lack of inhibition by clinically used carboplatin-etoposide combination seen in this synergy experiment point to any issues with the other conclusions drawn from this in vitro work? (line 414)

We thank the reviewer for the comment. The limited synergy of carboplatin–etoposide does not affect our conclusions. This combination is known to be schedule- and exposure-dependent, showing maximal efficacy when carboplatin precedes etoposide and after repeated dosing, unlike our single-time, concurrent low-dose setup(4,5). These factors explain the weak in vitro response, which is consistent with prior reports and does not affect the validity of our findings, as the same assay reliably identified synergistic carboplatin–gemcitabine effects that translated in vivo. The following changes were made:

Line 422-425: “This combination is known for its schedule- and exposure-dependent activity, achieving maximal efficacy with sequential (carboplatin-before-etoposide) and repeated dosing rather than the single-time, concurrent low-dose exposure used in our assay(40,41).”

7. Figure legends: please include statistical test, n, and indications of what asterisks mean

We thank the reviewer for this comment. The requested information was added to the legends of the figures.

8. Figure 3 title is an overstatement – the EdU analysis does not show “mechanism of action”. Please rephrase.

The Fig. 3 was renamed: “Effects of gemcitabine and carboplatin on DNA synthesis”.

9. The authors use a variable slope model to analyse their dose response curves. How many data points did they have between 0.78 to 100uM (how many are needed to use variable slope vs standard slope?)

We thank the reviewer for this question. There were 8 points in the dose-response studies, which is within GraphPad Prism’s recommended 6–10 points for reliable variable-slope (four-parameter) fitting(6). The additional slope parameter was retained as residual analysis confirmed improved fit over the fixed-slope model.

10. Please mention the post hoc test used for one way ANOVA (line 250)

We thank the reviewer for this comment. The Dunnett test was used for two-way ANOVA in this case. This information was added to the figure legend.

11. It would be helpful if it were possible to include tumor/eye images for the in vivo study

We thank the reviewer for this comment. The representative images were added as S7 Fig.

References

1. Kano Y, Akutsu M, Suzuki K, Yoshida M. Effects of carboplatin in combination with other anticancer agents on human leukemia cell lines. Leuk Res. 1993 Feb 1;17(2):113–9.

2. Rowinsky EK, Citardi MJ, Noe DA, Donehower RC. Sequence-dependent cytotoxic effects due to combinations of cisplatin and the antimicrotubule agents taxol and vincristine. J Cancer Res Clin Oncol. 1993 Dec;119(12):727–33.

3. Lee K, Tanaka M, Kanamaru H, Hashimura T, Yamamoto I, Konishi J, et al. In vitro antagonism between cisplatin and vinca alkaloids. Br J Cancer. 1989 Jan;59(1):36–41.

4. Hande KR. The Importance of Drug Scheduling in Cancer Chemotherapy: Etoposide as an Example. STEM CELLS. 1996;14(1):18–24.

5. Meczes EL, Pearson ADJ, Austin CA, Tilby MJ. Schedule-dependent response of neuroblastoma cell lines to combinations of etoposide and cisplatin. Br J Cancer. 2002 Feb;86(3):485–9.

6. Motulsky H, Christopoulos A. Fitting models to biological data using linear and nonlinear regression A practical guide to curve fitting. [Internet]. Oxford University Press; 2004 [cited 2025 Oct 30]. Available from: https://www.scirp.org/reference/referencespapers?referenceid=896670

---

## [Decision Letter · Decision Letter 3]

7 Dec 2025

High-throughput screening of drug libraries identifies a new synergistic drug combination for the treatment of retinoblastoma

PONE-D-24-22008R3

Dear Dr. Dyson,

We’re pleased to inform you that your manuscript has been judged scientifically suitable for publication and will be formally accepted for publication once it meets all outstanding technical requirements. The authors perfectly addressed all reviewers' comments.

Kind regards,

Irina V. Lebedeva, Ph.D.

Academic Editor

PLOS One

Additional Editor Comments (optional):

Reviewers' comments:

Reviewer's Responses to Questions

**Comments to the Author**

Reviewer #6: All comments have been addressed

Reviewer #7: All comments have been addressed

Reviewer #8: All comments have been addressed

2. Is the manuscript technically sound, and do the data support the conclusions?

Reviewer #6: (No Response)

Reviewer #7: Yes

Reviewer #8: Yes

3. Has the statistical analysis been performed appropriately and rigorously?

Reviewer #6: (No Response)

Reviewer #7: Yes

Reviewer #8: Yes

4. Have the authors made all data underlying the findings in their manuscript fully available?

Reviewer #6: (No Response)

Reviewer #7: Yes

Reviewer #8: Yes

5. Is the manuscript presented in an intelligible fashion and written in standard English?

Reviewer #6: (No Response)

Reviewer #7: Yes

Reviewer #8: Yes

Reviewer #6: (No Response)

Reviewer #7: The article has already been extensively reviewed by multiple reviewers. The authors seem to have addressed all the queries raised. i believe the paper is ready for publication

Reviewer #8: The authors have effectively addressed all reviewer concerns, although the eye survival method details requested by reviewer 6 do not appear to be in the methods and should be added, as the authors mention in their Response to Reviewers.

**Do you want your identity to be public for this peer review?** For information about this choice, including consent withdrawal, please see our Privacy Policy

Reviewer #6: No

Reviewer #7: No

Reviewer #8: No

---

## [Editor Report · Acceptance letter]

PONE-D-24-22008R3

PLOS One

Dear Dr. Dyson,

I'm pleased to inform you that your manuscript has been deemed suitable for publication in PLOS One. Congratulations! Your manuscript is now being handed over to our production team.

Kind regards,

on behalf of

Dr. Irina V. Lebedeva

Academic Editor

PLOS One